

# Isotopic evaluation of the National Water Model reveals missing agricultural irrigation contributions to streamflow across the western United States

Annie L. Putman[1], Patrick C. Longley[2], Morgan C. McDonnell[1], James Reddy[3], Michelle Katoski[4], Olivia L. Miller[1], and J. Renée Brooks[5]

[1]US Geological Survey Utah Water Science Center
[2]US Geological Survey Colorado Water Science Center
[3]US Geological Survey New York Water Science Center
[4]US Geological Survey Maryland-Delaware Water Science Center
[5]US Environmental Protection Agency, Pacific Ecological Systems Division

**Correspondence:** Annie L. Putman (aputman@usgs.gov)

**Abstract.** The National Water Model (NWM) provides critical analyses and projections of streamflow that support water management decisions. However, the NWM performs poorly in lower elevation rivers of the western United States (US). The accuracy of the NWM depends on the fidelity of the model inputs and the representation and calibration of model processes and water sources. To evaluate the NWM, we performed a water isotope ($\delta^{18}O$ and $\delta^{2}H$) mass balance using long term mean

summer hydrologic fluxes between 2000 and 2019, and gridded precipitation and groundwater isotope ratios. We compared the NWM-flux-estimated ('model') river reach isotope ratios to 4503 in-stream water isotope observations in 877 reaches across 5 basins in the western US. A simple regression between observed and mass balance estimated isotope ratios explained 57.9% ($\delta^{18}O$) and 67.1% ($\delta^{2}H$) of variance, though observations were 0.5‰ ($\delta^{18}O$) and 4.8‰ ($\delta^{2}H$) higher, on average, than mass balance estimates. The unexplained variance suggest that the NWM does not include all relevant water fluxes to rivers. To infer

possible missing water fluxes, we evaluated patterns in observation-model differences using $\delta^{18}O_{diff}$ ($\delta^{18}O_{obs} - \delta^{18}O_{mod}$) and $d_{diff}$ ($\delta^{2}H_{diff} - 8*\delta^{18}O_{diff}$). We detected evapoconcentration of observations relative to model estimates (negative $d_{diff}$ and positive $\delta^{18}O_{diff}$) at lower elevation, higher stream order, arid sites. The catchment actual evaporation to precipitation ratio, the fraction of streamflow estimated to be derived from agricultural irrigation, and whether a site was reservoir-affected were all significant predictors of $d_{diff}$ in a linear mixed effects model, with up to 15.1% of variance explained by fixed effects.

This finding is supported by patterns in groundwater levels and groundwater isotope ratios, and suggests the importance of including irrigation return flows to rivers, especially in lower elevation, higher stream order, arid rivers of the Western US.

## 1 Introduction

The western United States (US) is experiencing multidecadal drought (Williams et al., 2022) and declining streamflows (Milly and Dunne, 2020). Major rivers are running dry (Kornfield, 2022), lakes are shrinking (Ramirez, 2022; Fergus et al., 2020, 2022),

and water users are experiencing shortages and cuts (Bureau of Reclamation, Department of the Interior, 2022). These de-





creases in streamflow and groundwater fluxes are projected to continue in coming years (Miller et al., 2021b, a), with projected decreases in snowpack (Mote et al., 2021; Siirila-Woodburn et al., 2021) and increases in temperatures (Hicke et al., 2022). Under drought and snow drought stress, as well as changing wintertime precipitation patterns, river flows may become more difficult to forecast (Hammond and Kampf, 2020; Siirila-Woodburn et al., 2021). Yet, with decreasing water availability, water

managers and other stakeholders tasked with managing and responding to current and future water supply increasingly depend on accurate streamflow predictions.

Fully routed, high spatial and temporal resolution streamflow models, like the National Oceanic and Atmospheric Administration's National Water Model (NWM) which utilizes Weather Research and Forecasting (WRF) Hydro model data (Gochis et al., 2018), provide short and medium term streamflow prediction in the United States, as well as analyses of past stream dis-

charge at ungaged locations. The accurate, detailed, frequent results from the National Water Model may be used by emergency managers, reservoir operators, floodplain managers, and farmers to aid in water use decision making and flood or pollution risk evaluation. The accuracy of predictions and current snapshots produced by the model depend on 1) inclusion and faithful representation of relevant water sources and hydrologic processes 2) appropriate calibration of parameter estimations and 2) the fidelity of the model inputs.

With respect to the faithful representation of water sources, the major water sources to streams in the mountainous west include two broad water flux categories: runoff (also called quickflow) and groundwater discharge (also called baseflow). Runoff during the summer comes from late season snowmelt, rain, and irrigation water. Groundwater discharge comes from shallow or deep in-ground water, typically recharged at high elevation by snowmelt. Rivers in the west derive the majority of their water from springtime melt of high elevation wintertime snowpack (Li et al., 2017; Hammond et al., 2023) and little water

is contributed to streams at lower elevations where there is minimal snowpack (Miller et al., 2021b). Some of the melt water enters streams as runoff during late spring and summer, while the remainder recharges shallow and deep groundwater and later in the season or in subsequent years enters the stream as groundwater discharge (Barnhart et al., 2016; Miller et al., 2021a; Brooks et al., 2021; Wolf et al., 2023). Rain contributes runoff to streamflow, but even in areas receiving a substantial proportion of their total annual precipitation during summer in association with the North American Monsoon, only a small proportion

of the total precipitation makes it to the stream  (Solder and Beisner, 2020; Tulley-Cordova et al., 2021); the remainder is evaporated from soils or transpired by plants (Milly and Dunne, 2020). Thus, lower elevation streams, particularly later in the summer, depend heavily on groundwater discharge from higher elevations to sustain their flows (Miller et al., 2016) and the majority of streams in lower elevation arid areas are likely to lose water to shallow groundwater recharge (Jasechko et al., 2021).

Within this hydrologic framework, human water use and management introduces complexity via reservoirs and managed release schedules, trans- and interbasin transfers, conveyances, and surface and groundwater withdrawals, as well as irrigation for agricultural crop or turf grass growth. Turf irrigation in cities composes the majority of household water use in most municipalities and agricultural irrigation can comprise up to 80% of total statewide water use in Western US states (Dieter et al., 2018). Water used for agricultural crop or turf grass growth locally intensifies water balance fluxes, through increases in

both water application and evapotranspiration in these select tracts of land. Depending on the method, both agriculture and turf





grass irrigation can contribute to local groundwater recharge (Grafton et al., 2018), with greater recharge coming from flood irrigation compared to sprinkler or drip irrigation methods. Water for irrigation can come from either surface or groundwater withdrawals. The irrigation water source may have both direct and indirect influences on streamflows particularly during low flow seasons (Essaid and Caldwell, 2017; Condon and Maxwell, 2019). However, these processes and fluxes are not currently
explicitly included in the NWM.

    Past NWM evaluations have leveraged streamgage measurements (Hansen et al., 2019; Towler et al., 2023; Seo et al., 2021) and model evaluation using streamgage measurements is included in the NWM WRF-Hydro workflow (Gochis et al., 2018). Using measured discharge to evaluate the NWM is useful because the data are publicly available at high spatial and temporal resolution (e.g., dataset used in  (Towler et al., 2023)). However, evaluation of streamflows with measured discharge 1) may
allow modelers to get the correct total streamflow values and temporal patterns at a reach for the wrong process reasons or 2) may suggest that the model could be improved due to mismatches between measured and modeled data, but cannot provide information on the specific process(es) or sources responsible for the errors.

    Among the climatic regions covered by the NWM, model streamflow evaluation metrics perform the most poorly in the Western US in lower elevation reaches. Metrics like the Kling–Gupta efficiency (KGE) indicate pervasive mismatches between
measured and modeled streamflows and percent bias (PBIAS) results showed that simulated streamflow volumes tend to be overestimated in the west  (Towler et al., 2023). Similarly, Hansen et al. (2019) found that the NWM has difficulty estimating flows during drought or low flow years in the Colorado River Basin. In the low elevation stream reaches of the Western US, disagreement between the NWM flows and observations within anthropogenically-altered reaches may come from incomplete representation of anthropogenic water sources or processes in the NWM.

75        In the western US, low elevation waterways have moderate to high potential for anthropogenic alteration (Fergus et al., 2021). For example, rivers and surface water supplies are managed by dams, and a large proportion of total water use is allocated to irrigating agriculture (Dieter et al., 2018). However, the NWM does not explicitly include surface water removal for agricultural irrigation nor subsurface return flows from irrigation in its streamflow computations. Likewise, the NWM represents inflow and outflow of lakes and reservoirs as passive storage and releases, with no active reservoir management. Both
of these omissions may be contributors to the large errors observed in the NWM in lower elevation areas where land use includes large amounts of along-river agriculture and streamflow is heavily managed through reservoir operations. Unfortunately, the effects of contributions of these two water sources on streamflow are difficult to identify and quantify through evaluations of streamflow records alone.

    Tracers, including the stable isotopes of water ($^{16}O$, $^{18}O$ and $^{1}H$, $^{2}H$), can be used to track the contributions of specific
water sources (e.g., groundwater fluxes, runoff fluxes) that may be impossible to disentangle from comparisons with total or instantaneous discharge. Stable isotopes of oxygen and hydrogen (expressed in delta notation, $\delta = 1000 * \left( \frac{R_{sample} - R_{standard}}{R_{standard}} \right)$ , where $R$ is the ratio of concentrations of the heavy ($^{2}H$ and $^{18}O$) and light ($^{1}H$ and $^{16}O$) isotopes) have diagnosed process limitations in other modeling contexts (Nusbaumer et al., 2017; Putman et al., 2019), leveraging the fractionation occurring for stable isotopes of waters during phase changes (Bowen et al., 2019). The secondary parameter, deuterium excess (defined
as $d = \delta^{2}H - 8 * \delta^{18}O$), is a reliable metric for detecting evaporation of precipitation and surface waters, evaporation under





a vapor pressure gradient or non-equilibrium condensation processes, like snow formation in mixed phase clouds or isotopic fractionation during the melting of snow (Putman et al., 2019; Bowen et al., 2018; Ala-aho et al., 2017).

Because hydrologic processes including groundwater recharge, discharge, and precipitation runoff do not cause isotopic fractionation, we can use water fluxes from hydrologic models with estimates of the isotope ratios of those fluxes on the
appropriate timescales to produce river water isotope estimates. This works well because the groundwater and runoff fluxes to summertime streamflow in the Western US have different, and often distinct, stable isotope ratios due to seasonal and spatial controls on precipitation isotope ratios. The signatures of groundwater inflow and snowmelt tend to have the lowest isotope ratios of the water sources in the hydrologic system and tend to be relatively temporally invariant (Bowen, 2008; Feng et al., 2009; Jasechko et al., 2014; Solder and Beisner, 2020; Tulley-Cordova et al., 2021). In contrast, summer precipitation, which
contributes runoff to streams, tends to have higher isotope ratios than groundwater (Jasechko et al., 2014; Tulley-Cordova et al., 2021).

Anthropogenic modifiers of streamflow that are not included explicitly in the NWM may be expected to alter the isotopic signature of streamflow downstream of the river water source areas in the headwaters. Agricultural irrigation can contribute both runoff to streams and recharge groundwater (Essaid and Caldwell, 2017; Gochis et al., 2018). This specific water source
becomes isotopically enriched during conveyance and application both due to direct evaporation during conveyance and soil evaporation after application (Craig and Gordon, 1965; Yang et al., 2019). The evaporated isotope signatures of agricultural soil waters are also evident in fruit water (Oerter et al., 2017). Thus, irrigation return flows (runoff or ground) associated with irrigation are much more evaporatively enriched, with lower $d$, than naturally recharged groundwater or precipitation runoff. The effects of evaporation on the isotope ratios of the return flows are expected to be greater in arid areas with higher
summer temperatures and higher vapor pressure deficits. Although lakes can be evapoconcentrated relative to other surface waters (Bowen et al., 2018), we do not expect similar signals of evaporative enrichment from reservoirs. Relative to natural lakes across the US, evaporation rates from western lakes are low relative to inflow (Brooks et al., 2014). Instead, reservoirs may alter the isotope ratios of streamflow through retention of and later discharge of spring snowmelt. Thus, reservoir outflow may have lower isotope ratios and higher $d$ than the upstream rivers during the summer months.

In this study, we performed an isotope mass balance using the water fluxes of the NWM and gridded water stable isotope datasets. We compared the isotope mass balance-derived long term mean streamflow isotope ratios directly with stream water isotope observations. If the NWM constrains all water sources affecting streamflow, we expect the differences between the isotope mass balance results and isotopic observations (observation-model differences) will be small and be uniformly positive or negative throughout each basin. If we observe spatial and/or seasonal variability and structured patterns in observation-
model differences within basins (i.e., patterns with elevation, stream order, or aridity), particularly with respect to the sign of the difference, we may infer that the NWM is incorrectly partitioning runoff and groundwater fluxes, or missing important water sources. We hypothesize that if we observe spatial variability and structured patterns in our observation-model difference data, we will observe higher isotope ratios and lower $d$ in more arid reaches reflecting the influence of evapoconcentrated irrigation return flows on streamflow as compared to higher elevation, humid or seasonally snowy reaches with minimal anthropogenic
influence.





## 2  Methods

Using an isotope mass balance approach, we calculated the long term mean stable isotope ratios of river reaches in the western US (Figure 1). Water fluxes were supplied by NWM simulations of groundwater and runoff fluxes (National Oceanographic and Atmospheric Administration, 2022) and isotope ratios came from gridded groundwater and precipitation stable isotope
products (Bowen, 2022b; Bowen et al., 2022). We compared calculated river isotope ratios to observations of isotope ratios. The data compiled and calculated data presented and analyzed in this manuscript are publicly available (Reddy et al., 2023). Differences between observations and modeled data were compared in an error-partitioning framework. Where applicable, results were interpreted in terms of the estimates of the influence of agricultural water use in the upstream area as well as the presence of impoundments on the river. A groundwater isotope ratio dataset and a well water surface elevation relative to river
surface elevation dataset from Jasechko et al. (2021) were used as independent lines of evidence supporting our analysis of observation- mass balance estimate differences.

### 2.1  Temporal domain

Our analysis was constrained to summer months (June, July, August) between 2000 and 2019. The specific months chosen reflect those with greatest evapotranspiration and thus consumptive water use and correspond to the season with the largest
number of spatially distributed river water isotope observations.

### 2.2  Spatial domain

We selected 5 HUC2 basins (U.S. Geological Survey, National Geospatial Technical Operations Center, 2023) in the Western US to compose our study area: the Upper Colorado (14), Lower Colorado (15), Great Basin (16), Pacific Northwest (17), and California (18). All basins were characterized by rivers sustained by wintertime snowpack mediated by groundwater infiltration
and discharge. All basins also included water management through impoundments and substantial water use for agriculture. In a simplified Köppen climate classification (Rubel and Kottek, 2010), the southern and central portions of the study area were characterized as arid, whereas much of the northern and mountainous portions of the study area was classified as warm temperate or seasonally snowy.

### 2.3  Data assimilation framework

Various data types used in this analysis (point, raster, vector) were spatially joined to coarsened catchments (n=15787) derived from the National Hydrography Dataset Plus (NHDPlus, (U.S. Geological Survey, 2019), see Text S1 for details on catchment coarsening). The full NHDPlus network was clipped to the spatial domain of our study. Each catchment contained one reach. Coarsened catchments had a median size of 51 km$^2$ and a mean size of 221 km$^2$, and flowlines had a median length of 20 km$^2$ and a mean length of 32 km$^2$. We also utilized some attributes provided with the NHDPlus, including the catchment area,
Strahler stream order, the reach length, minimum and maximum catchment elevation, and the feature code, which denoted the flowline path type.

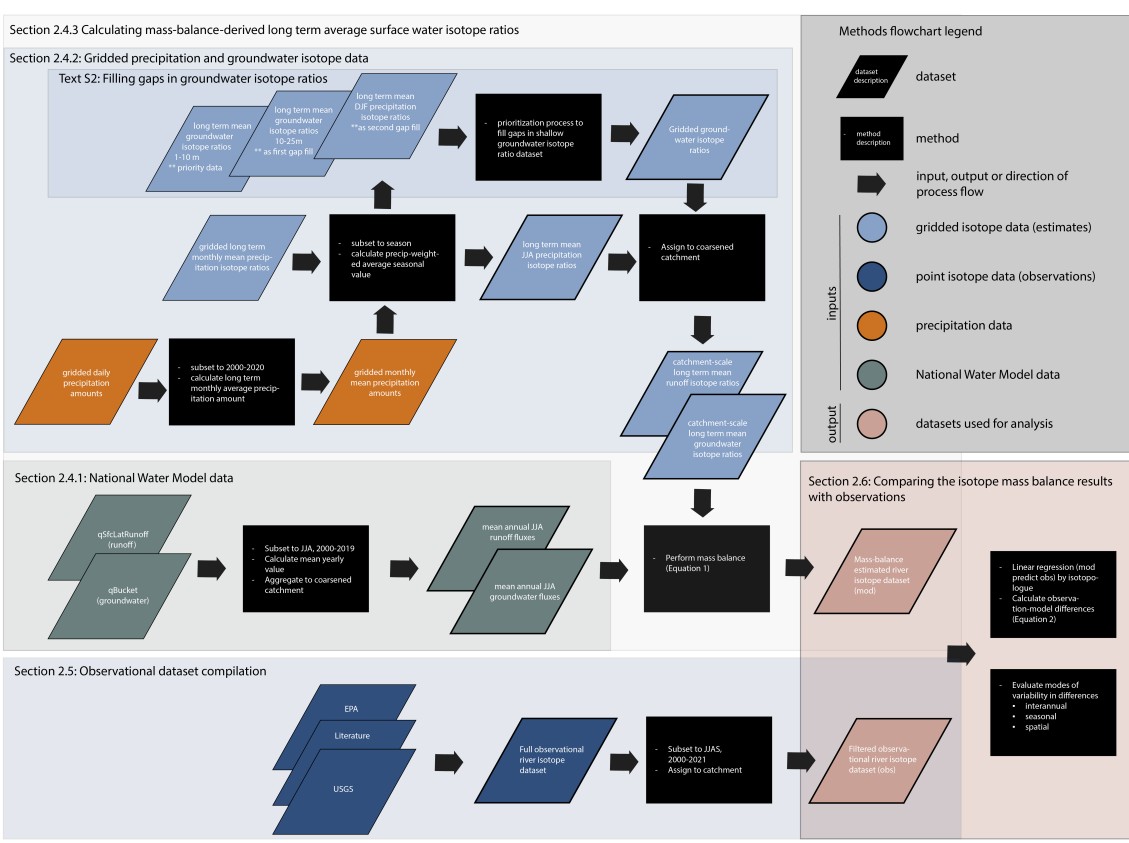

**Figure 1.** Diagram showing methods and datasets as described in Sections 2.4-2.6. Four datastreams were used to formulate the long isotope mass balance estimates of river isotope ratios: gridded precipitation isotope estimates Bowen (2022b), gridded groundwater isotope estimates (Bowen et al., 2022), gridded precipitation data (University of East Anglia Climatic Research Unit and Harris, I.C. and Jones, P.D. and Osborn, T. , 2021) and NWM data (National Oceanographic and Atmospheric Administration, 2022) . Three data categories contributed to the observational river isotope dataset: USGS (U.S. Geological Survey, 2022), EPA (U.S. Environmental Protection Agency, 2016b, 2020), and literature datasets accessed from the waterisotopesdb.org  (Putman and Bowen, 2019).





## 2.4 Performing the isotope mass balance with NWM water fluxes and gridded water isotope ratios

Using estimates of long term mean groundwater and precipitation isotope ratios (Bowen, 2022b; Bowen et al., 2022), we applied an isotope mass-balance to the NWM groundwater and runoff fluxes to streams (Figure 1). The operational hydrologic

model is based on inputs from WRF-Hydro (Gochis et al., 2020b, a) and simulates and forecasts major water components (e.g., evapotranspiration, snow, soil moisture, groundwater, surface inundation, reservoirs, and streamflow) in real time across the CONUS, Hawaii, Puerto Rico, and the US Virgin Islands. In the NWM framework, surface and soil evaporation are wrapped into the evapotranspiration flux variable, and direct evaporation from rivers and reservoirs are not considered in the NWM surface water balance. Thus, we did not apply any additional isotopic fractionation to the groundwater and runoff isotopic

fluxes. This approach produced the 'characteristic' long term mean isotope ratio expected for river reaches in the western US. These estimates were directly comparable to river water isotope observations.

### 2.4.1 National Water Model data

We accessed the groundwater 'bucket' (qBucket, $m^3$ $s^{-1}$) runoff (qSfcLatRunoff, $m^3$ $s^{-1}$), and streamflow (streamflow, $m^3$ $s^{-1}$) fluxes from the NWM v 2.1 Analysis Assimilation dataset (National Oceanographic and Atmospheric Administration,

2022) . We used runoff and groundwater fluxes for our mass balance estimates (Figure 2), and retained streamflow as a reach scale quantity to be included in data analyses. These variables are available at the NHDPlus reach scale on an hourly timestep between 2000 and 2019. We subset these variables to the summer months (June, July, and August) and calculated the mean water fluxes to each reach for the summer season of each year. The interannual variability in the summer fluxes was leveraged as an estimate of the uncertainty of the long term mean summer water fluxes.

### 2.4.2 Gridded precipitation and groundwater isotope data

The precipitation and groundwater stable isotope ratios ($\delta^2$H, $\delta^{18}$O) that we used to perform the isotope mass balance came from two different publicly available gridded products. Both represent long term means or climatologies and provide estimates of uncertainty.

We obtained monthly precipitation isotope ratio climatological predictions and uncertainty estimates (1 standard deviation)

for both H and O from Bowen (2022b). The monthly USA grids were available at 1 km, and were produced with the OIPC v3.2 database (Bowen, 2022a) following methods described in Bowen et al. (2005). Monthly grids have been adjusted for consistency with annual values (see version notes for OIPC2.0 (Bowen, 2006)). We calculated the precipitation-weighted long term mean summer (June, July, August) and winter (December, January, February) seasonal isotope ratio climatologies with long term monthly mean precipitation climatologies calculated from the Climatic Research Unit (CRU) mean monthly

precipitation amounts (Harris et al., 2020; University of East Anglia Climatic Research Unit and Harris, I.C. and Jones, P.D. and Osborn, T. , 2021) for the period 2000-2020. The precipitation weighted mean seasonal climatology error was calculated analytically from the timeseries.



The groundwater isoscapes used in this analysis were produced by Bowen et al. (2022) for 7 depth intervals ranging from 1 to 1000m. The groundwater isoscapes were not temporally resolved. Because this project focuses on groundwater discharge to streams, we preferentially utilized the 1-10m depth interval. However, this layer contained some data gaps where insufficient well data were present to perform an estimate. Where available, we filled these data gaps using method outlined in Text S2. The groundwater isotope ratio data included estimates of uncertainty, which were retained for the characterization of uncertainty around the mass balance isotope ratio estimates.

The gridded precipitation and groundwater isotope datasets and their uncertainties were assimilated to the NHDPlus spatial framework. Because the raster data grid sizes were larger than the catchment sizes we employed a distance minimization approach using the centroid of the catchment and the centroids of the grid cells.

### 2.4.3 Calculating mass-balance-derived long term mean surface water isotope ratios

To estimate the long term mean surface water isotope ratio ($R_{sw,r}$) at each reach ($r$) in the spatial domain (Equation 1), we accumulated the groundwater ($gw$) and runoff ($ro$) isotope fluxes (i.e., the isotope ratio multiplied by the water flux, $R*F$) for all reaches ($i$) from the headwaters downstream to the reach. The isotope ratio for runoff ($R_{ro}$) came from the summer mean gridded precipitation isotope ratios and the isotope ratio for the groundwater flux ($R_{gw}$) came from the gridded groundwater isotope ratios (see Section 2.4.2). The summed isotope fluxes were divided by the summed runoff and groundwater fluxes.

$$R_{sw,r} = \frac{\sum_{i=0}^{r} R_{gw,i} * F_{gw,i} + R_{ro,i} * F_{ro,i}}{\sum_{i=0}^{r} F_{gw,i} + F_{ro,i}} \tag{1}$$

Because volumetric contributions of groundwater and runoff to streamflow in each reach varied with year, and because the long term mean estimates of the groundwater and precipitation isotope ratios include estimates of uncertainty, we evaluated the potential for uncertainty in our estimates of $R_{sw,r}$. We used the interannual variability in the mean summer water fluxes as characteristic of the uncertainty in our estimate of our long term mean water fluxes, and used the uncertainty estimates provided with the isotope ratio products as representative of the uncertainty in the isotope ratios. We performed 10 random draws of the isotope ratio distributions for each of the 20 years of record, which gave a total of 10 estimates of $R_{sw}$ per year over 20 years of record, or 200 total estimates of $R_{sw}$ per reach. Joint distributions (of either H and O, or isotopes with water fluxes), were not used because information about how the isotope ratios might covary was not available from the gridded isotope datasets and no assumptions were made about how the isotopes might vary with interannual variability in climatic conditions. Similarly, no assumptions were made that the precipitation and groundwater isotope ratios covaried in time. These 200 estimates were used to calculate a long-term mean estimated isotope ratio for river water in each reach of the network and to evaluate uncertainty in our estimates.

### 2.5 Compilation of river isotope observations

The results of the mass balance were compared with observations of stable water isotope ratios from rivers collected between 2000 and 2021, during the growing season months of June, July, August and September. We included two additional years (2020 and 2021) as well as data from the month of September beyond the temporal constrains of the NWM model domain in



our set of observations. This decision as made to maximize the amount of data and number of unique river reaches in the spatial domain that are available for analysis, and reflects the assumption that the long term mean river isotope ratios calculated from the mass balance approach will be insensitive to inclusion or exclusion of a small number of additional years or an additional growing season month.

We compiled surface water stable isotope ($\delta^2$H, $\delta^{18}$O) measurements from various sources including the Environmental

Protection Agency (EPA), the United States Geological Survey (USGS) National Water Information System (NWIS, U.S. Geological Survey (2022)), and published datasets assimilated in the Water Isotopes Database (Putman and Bowen, 2019). Not all reaches had one or more stable water isotope observations, and river reaches with multiple stable water isotope ratio observations were sometimes, but not always, from the same sampling site within the catchment.

The EPA surface water stable isotope data came from the National Rivers and Streams Assessments (NRSA, U.S. Environ-

mental Protection Agency (2016b, 2020)) and the National Lakes Assessment (NLA, U.S. Environmental Protection Agency (2009, 2016a)). These data were collected once or twice per summer on a five year rotating basis as part of routine sampling campaigns. Over the time period of our analysis, we obtained three collections of NRSA samples (2008-2009, 2013-2014, and 2018-2019). Sites were sometimes but not always resampled among the campaigns. Sampling was stratified based on Strahler stream order and by state ensuring that all orders were sampled within each state in the assessments (U.S. Environmental

Protection Agency, 2016b, 2020). This means that higher order reaches are less frequently sampled than medium or low order reaches.

The USGS surface water stable isotope data for rivers were downloaded through the NWIS API (U.S. Geological Survey, 2022) and the literature data came from published and unpublished sources that are publicly available through the Water Isotopes Database (Putman and Bowen, 2019). Stable isotope collections are not part of routine measurements for the USGS,

but rather are collected by specific USGS projects. Thus, stable isotope data collections from the USGS and literature datasets tended to be spatially and temporally clustered.

### 2.6 Comparing the isotope mass balance results with observations

The relationship of the NWM isotope mass balance (modeled) to the river isotope observations were evaluated using correlation and simple regression analyses, where the modeled isotope ratio (either $\delta^2H$ or $\delta^{18}O$) values are used to predict the observed

isotope ratios. We evaluated the results with all unaveraged observations and mean isotope ratio at river reaches with multiple observations. A Pearson correlation analysis was performed using the 'corr()' function of python's 'pandas' package (Wes McKinney, 2010; The pandas development team, 2020). Regression analysis was performed using the ordinary least squares (OLS) function in the python 'statsmodels' package (Seabold and Perktold, 2010).

We calculated the likelihood that an observation and the model result came from the same distribution, based on the variance

in the model estimate, and variance associated with river water isotope observations ( Text S3) using a two-tailed t-test. We reported p-values, where p<0.1 indicated that the isotope mass balance estimate was statistically different from the observed surface water isotope ratio for the specific element (H or O).





### 2.6.1 Calculating observation-model differences

We calculated the observation ($obs$)-model estimate ($mod$) differences in both $\delta^{18}O$ and $\delta^2H$, by subtracting the model estimate from the observation ($\delta^{18}O_{diff} = \delta^{18}O_{obs} - \delta^{18}O_{mod}$; $\delta^2H_{diff} = \delta^2H_{obs} - \delta^2H_{mod}$). Using both isotope systems, we established a framework for interpretation of our results (Figure 2) that utilizes movement along or deviation from the global mean $\delta^2H : \delta^{18}O$ ratio of 8 that is used to represent fractionation that occurs at equilibrium and defines the slope of the Global Meteoric Water Line (GMWL Craig, 1961).

Observation-model differences may arise from either 1) incorrect model source representation (i.e., missing water sources or incorrect fluxes of established sources) or 2) errors in the isotope ratio datasets used for the isotope mass balance calculation. Thus, for positive or negative values of $\delta^{18}O_{diff}$ and $\delta^2H_{diff}$ that exhibit a $\delta^2H_{diff} : \delta^{18}O_{diff}$ ratio of 8, we infer either errors in NWM with respect to the proportions of runoff and groundwater contributed, or errors in the gridded isotope ratios (likely groundwater, due to its disproportionate contributions). For positive or negative $\delta^{18}O_{diff}$ and $\delta^2H_{diff}$ with $\delta^2H_{diff} : \delta^{18}O_{diff}$ ratios different from 8, we infer that the NWM is missing uncharacterized water sources with isotope values bearing a signature of non-equilibrium fractionation. We quantify differences of $\delta^2H_{diff} : \delta^{18}O_{diff}$ ratios from 8 using a metric of similar to $d$ (Equation 2).

$$d_{diff} = \delta^2H_{diff} - 8 * \delta^{18}O_{diff} \tag{2}$$

We can interpret combinations of $\delta^{18}O_{diff}$ and $d_{diff}$ together, as well as $d_{diff}$ independently to infer the uncharacterized sources responsible for the observation-model difference. This framework is useful because the ratios of $\delta^2H$ to $\delta^{18}O$ of the isotopic inputs to the isotope mass balance tend to be close to 8 (Bowen, 2022b; Bowen et al., 2022) whereas those from the observations more often differ from 8 (U.S. Environmental Protection Agency, 2016b, 2020). This means that all non-zero $d_{diff}$ values can be used to identify omitted water sources and where they are important to streamflow.

### 2.7 Evaluating modes of variability in observation-model differences

We evaluated interannual, seasonal, and spatial modes of variability in our observation-model differences. The interannual variability was assessed to ensure that patterns in the other modes of variability did not arise due to the timescale difference between our isotope mass balance estimates (long term mean) and observations (instantaneous). The seasonal and spatial variability were investigated to understand the reasons for observation-model differences in our dataset.

### 2.7.1 Evaluating interannual variability in observation-model differences

To assess the influence of interannual variability on unexplained variance in our observation-model comparison, we utilized the years (2008, 2009, 2013, 2014, 2018, 2019) where sampling was conducted by the EPA NRSA, because these years are likely to have the most representative and consistent spatial distribution of samples. Every NRSA-sampled year had over 250 observations and represented more than 100 river reaches. No other years exhibited this spatial representation. We calculated

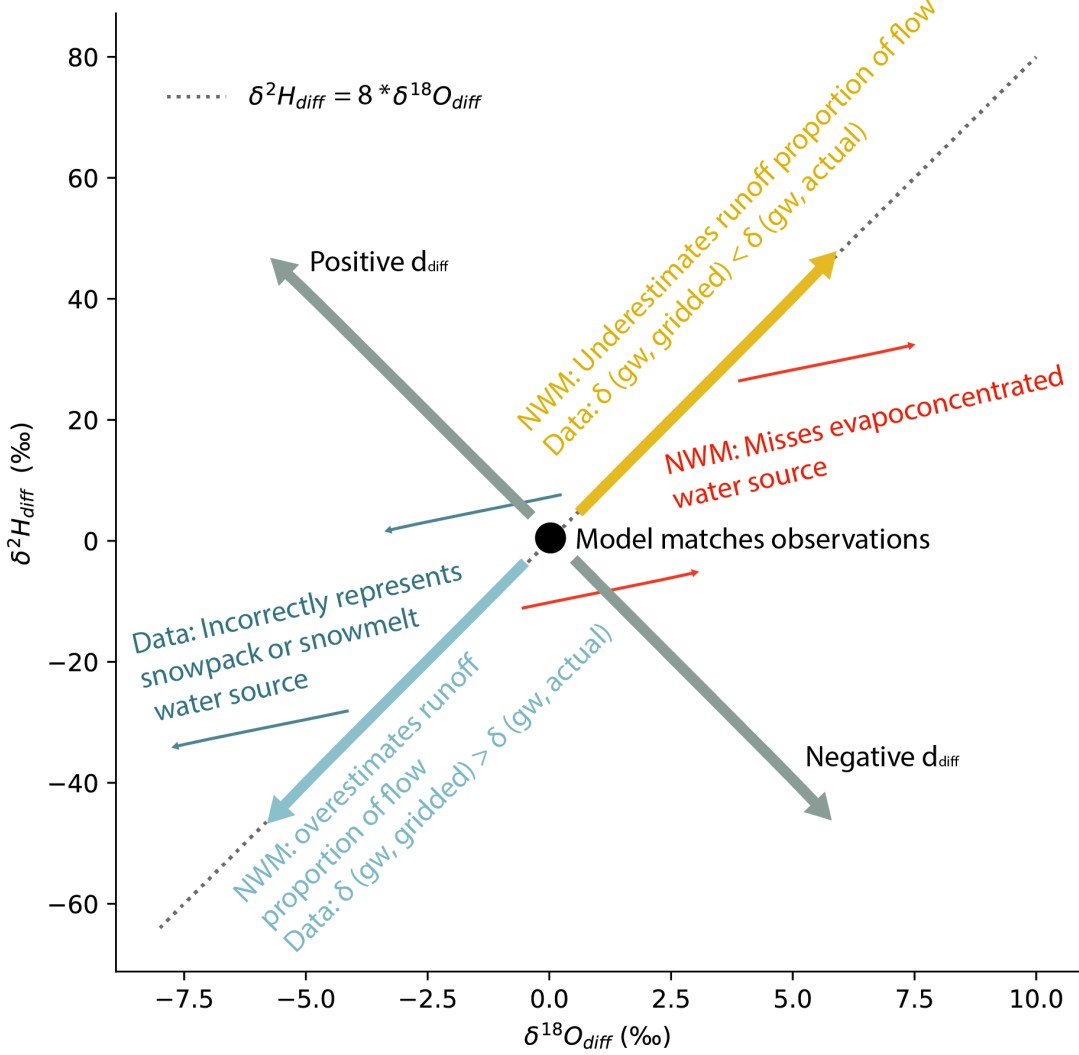

**Figure 2.** Schematic for interpretations of observation model differences utilizing dual isotope space and assumptions about the expected relationships between $\delta^{18}O_{diff}$ and $\delta^2H_{diff}$. The annotations associated with 'NWM' specify the sort of hydrologic model error (i.e. water source apportionment) that could produce the observation-model comparison result, if all isotope data supplied to the isotope mass balance are correct. The annotations associated with 'Data' specify the sort of error in the gridded isotope datasets that could produce the observation-model result if all NWM water source contributions are assumed to be correct.





the mean and standard deviation of the observation-model differences, and performed a regression on the differences (e.g., as described for all data in Section 2.6.1). The EPA method is sufficiently standardized (Theobald et al., 2007) that the regression
results for each year are unlikely to primarily reflect spatial variability in sampling locations.

### 2.7.2 Evaluating variability in observation-model differences across the growing season

To assess how the observation model difference may change over the growing season, we obtained all sites-year combinations where there were at least three observations during at least three of the four months (Jun-Sep) of the growing season. We required one of the months be the month of June to form the basis for comparison. From the June value(s) of $\delta^{18}O_{diff}$ and
$d_{diff}$ for a site-year combination, we subtracted the $\delta^{18}O_{diff}$ and $d_{diff}$ values calculated for other months at the same site and from the same year. We evaluated the distribution of the aggregate results, as well as the distributions at the HUC2 basin scale by comparing their means and interquantile ranges.

### 2.7.3 Evaluating sources of spatial variability in observation-model differences

Spatial structure in the observation-model differences were evaluated graphically by comparison of $\delta^{18}O_{diff}$ and $d_{diff}$ with
catchment mean elevation, Strahler stream order, and Köppen climate class (Rubel and Kottek, 2010). The former two variables were retained from the NHDPlus catchment dataset (U.S. Geological Survey, 2019). The former was joined to the spatial framework as described in Text S4.

Spatial structure in the observation-model differences were evaluated statistically with linear mixed effects modeling using the basin (HUC2) as a random variable using the python 'statsmodels' module and the 'mixedlm()' function (Seabold and
Perktold, 2010). Linear mixed effects modeling with basin as the random (grouping) variable was selected for the analysis method because water in streams at low elevations is likely to be more isotopically similar to water in the basin headwaters than a nearby stream in a different basin with different water source regions. Thus, we assume the groups are likely to have different mean values reflecting their hydrologic and climatic differences. Although we also expect that the relationship of the response variable $d_{diff}$ to the explanatory variables may differ among basins, both our response and explanatory variables
contain substantial scatter as well as small numbers of high leverage points in each basin, such that a more nuanced analysis would be likely to produce misleading results.

Using the linear mixed effects approach, we tested the statistical relationship between $d_{diff}$ and the ratio of actual evaporation to precipitation ($\frac{ET_a}{P}$, Text S4), catchment mean elevation, fraction of streamflow estimated to come from agricultural return flows (Text S5), and a categorical variable indicating influence of large reservoirs (capacity >50,000 acre-feet, Text
S5.2). We performed statistical analysis on all sites on streams not categorized as intermittent, ditches, or canals.

### 2.8 Evaluation of independent lines of evidence supporting signature of agricultural water use in rivers

Because it is difficult to disentangle the effects of elevation and aridity from the effects of human water use and management due to their spatial covariance, we utilized analyses of independent datasets to support the results of our statistical inference.





The analyses evaluated relationships between land use or cover and groundwater isotope ratios and the fraction of well water
levels that are below the nearby river level in catchments across the western US.

### 2.8.1 Associating groundwater stable isotope observations with land use / land cover types

Estimates of the isotopic evapoconcentration of groundwater associated with different land use and land cover classes supports
our inferences from observation-model differences. We made the associations between groundwater isotope ratios and land use
classes at a HUC12 scale (U.S. Geological Survey, National Geospatial Technical Operations Center, 2023).

We considered five land use type categories that were aggregations of two or more National Land Cover Database (De-
witz and U.S. Geological Survey, 2021) categories. The 'desert' category was composed of barren land (NLCD code=31),
shrub/scrub (52), and grasslands/herbaceous (71) land classes. The 'forest' category was composed of evergreen, deciduous
and mixed forests (41-43). The 'developed' category was composed of all the 'developed' classes, including open (21-24). The
'agriculture' category was composed of pasture/hay (81) and cultivated crops (82). The final category, 'water and wetlands'
included all other land types, which include open water (11), perennial ice/snow (12), woody wetlands (90) and emergent
herbaceous wetlands (95). We assigned the dominant land use/land cover category for each HUC12 using data based on the
land use type with the greatest fractional coverage.

We compiled groundwater stable isotope ($\delta^{18}O$, $\delta^2 H$) measurements from the USGS NWIS (U.S. Geological Survey, 2022),
and published datasets assimilated in the Water Isotopes Database (Putman and Bowen, 2019). We did not place temporal or
well depth constraints on the samples used in our analysis. Not imposing well depth constraints may contribute to scatter associ-
ated with differences in water sources recharging shallow groundwater compared to deeper confined aquifers. The groundwater
isotope ratio observations were spatially joined to the hydrologic units.

### 2.8.2 Evaluation of NWM groundwater discharge with well level fractions

The Jasechko et al. (2021) dataset compared river surface elevations with river-side well water elevations within catchments.
The approach produced the fraction of wells in a catchment whose water surface levels were lower than the water surface level
of the nearby river. In catchments where most well water levels are below the river water level (scores close to 1), we expect
the river to lose water to shallow groundwater recharge under the right geologic conditions (e.g., permeability). In contrast, in
catchments where most well water levels are above the river water level (scores close to 0), we expect groundwater discharge
to streams.

We predicted the long term mean summer NWM 'qBucket' magnitude using the Jasechko et al. (2021) dataset using a simple
linear regression. This approach tests the hypothesis that if NWM accurately represents groundwater discharge to streams, the
relationship of well water elevations to river surface elevation would predict the summer mean NWM groundwater discharge
flux (assuming a linear relationship between the two quantities), with some scatter to account for subsurface permeability and
spatial variability in groundwater discharge rates. We then evaluated the effect of agricultural irrigation in a catchment on the
relationship between NWM 'qBucket' (binned by to the 0-20$^{th}$, 20$^{th}$-40$^{th}$, 40$^{th}$-60$^{th}$, 60$^{th}$-80$^{th}$, and 80$^{th}$-100$^{th}$ percentiles)
and the Jasechko et al. (2021) dataset. The evaluation was split into reaches influenced by irrigation sourced from groundwater





and irrigation sourced from surface water, as well as reaches uninfluenced by irrigation water. Irrigation contributions and irrigation water sources were determined using the methods for estimating irrigation water use described in Text S5.1.

## 3 Results and discussion

### 350 3.1 Evaluation of the isotope mass balance approach for estimating surface water isotope ratios

The analysis evaluated 4503 stream stable isotope observations in 877 unique river reaches across the western United States relative to NWM-driven isotope mass-balance-derived estimates (hereafter, 'modeled') of the river isotope ratios. Of these, 448 reaches had more than one observation (often all at the same sampling site in the catchment, but sometimes at multiple sites, Figure S1), and up to 571 observations in a catchment (Figure S1 and S2). On average, across all data, the observations

were significantly greater than the modeled values by $0.537 \pm 0.033$ ‰ and $4.81 \pm 0.222$ ‰ for $\delta^{18}O$ and $\delta^2 H$, respectively (Figure 3). For $\delta^{18}O$ we observed a standard deviation of 3.16‰ for the observed data and 2.96‰ for the modeled data (for all data averaged by catchment). For $\delta^2 H$ we observed a sample standard deviation of 25.4‰ for the observed data and 24.4‰ for the modeled data (for all data averaged by catchment, Figure 3).

The modeled isotope ratios and observed isotope ratios were well correlated (Table 1, Figures S4-7), with correlation co-

efficients between 0.761 and 0.866, depending on the isotopologue and whether individual observations or catchment means were considered. These correlations translated to statistically significant simple linear regressions where the modeled isotope ratios were used to explain the observed isotope ratios (Table 1). Depending on the isotopologue and whether individual observations or means were considered, the model explained between $\sim 58\%$ and 75% of the variance in the observations. The model explained more variance for $\delta^2 H$ than $\delta^{18}O$, and more variance for catchment mean values relative to individual ob-

servations. For all regressions, the slopes ranged from 0.879 to 0.937, with catchment mean slopes tending to be lower than slopes calculated from all observations. Intercepts for all regressions were close to, but less than 0, with lower intercepts associated with regressions calculated from catchment mean values, relative to regressions calculated from all observations. The statistically significant slopes of less than 1 and statistically significant intercepts arise in all observation-model comparison regressions because the observations tended to exhibit higher isotope ratios than the model estimated at the lower end of the

isotopic distribution (Figures S4-7). Many of the catchments characterized by this pattern were in more arid locations.

We calculated surface water lines (SWL, similar to a meteoric water line (MWL), but instead of being calculated from precipitation, they are calculated with isotope ratios from surface water samples in an area) for both the modeled and observed results using all available data (Figure 3). The observations yielded a surface water line with a slope of 7.570 ($\pm 0.023$), and intercept of 1.2301 ($\pm 0.320$), which was significantly different from the slope of the GMWL slope of 8 and intercept of 10

and was within the range of local MWLs (LMWL) slopes for western North America (6.5-8) (Putman et al., 2019), reported in Table 2. The model results yielded a surface water line 8.12 ($\pm 0.010$) and an intercept of 8.06 ($\pm 0.14$) which was more similar to, but still statistically different from the GMWL and differed from LMWLs for the region (Table 2). Comparison of the observation and modeled data distributions and water lines reveals evidence for evapoconcentration of surface waters in the observations but not in the isotope mass balance results (Figure 3). This is because the primary source of streamflow in

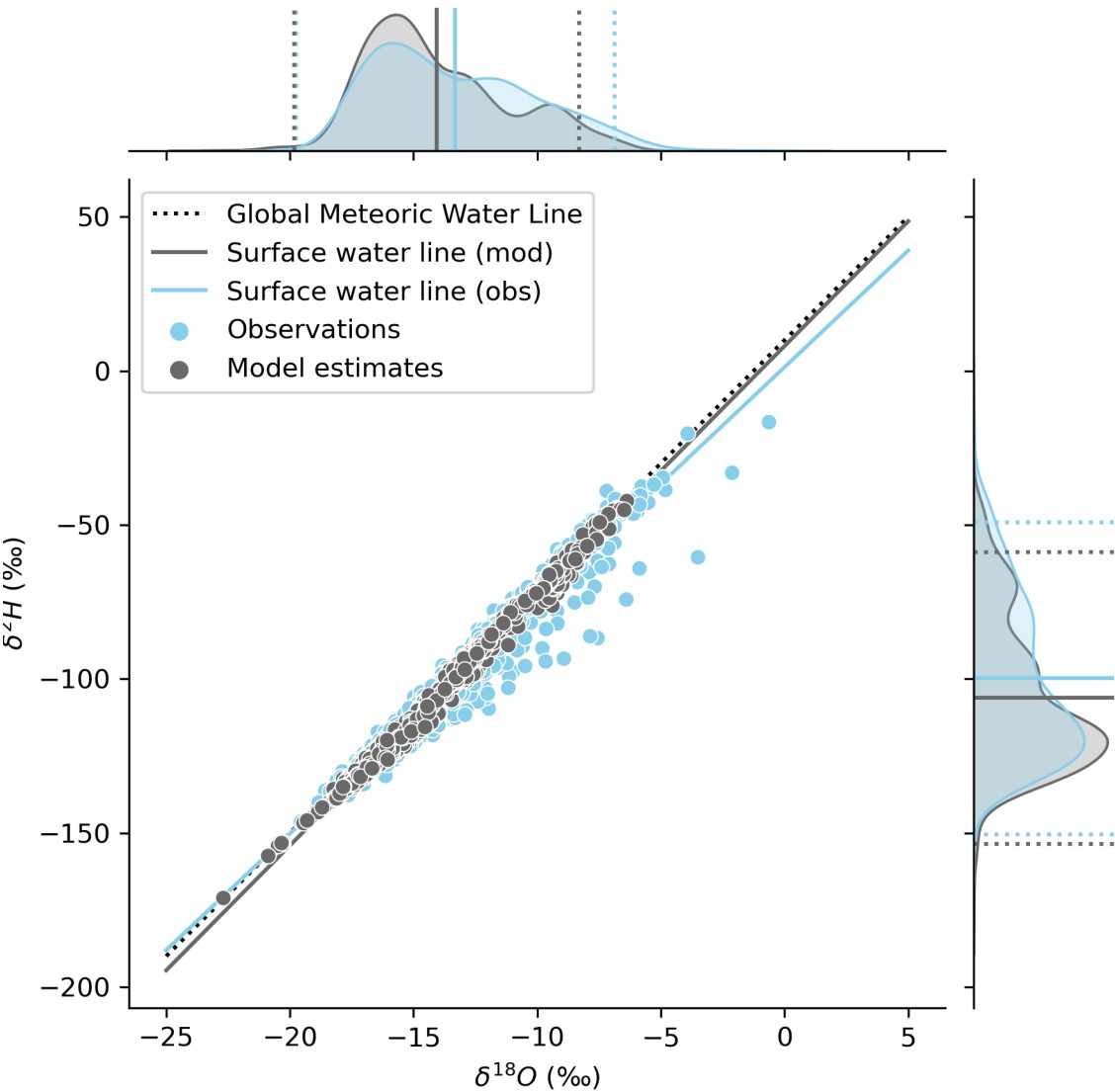

**Figure 3.** The distribution of the catchment mean observation (obs, blue) and isotope mass balance estimates (mod, gray) (n=448) with the Global Meteoric Water line (dotted) and the two datasets surface water lines (solid lines). Data distributions, including mean and two standard deviations of each data type (dotted lines), are shown in the plot margins. Observations plotting below the GMWL indicate evapoconcentration (Putman et al., 2019).





| Statistical model | n | Corr. coef | $\beta$ ($\pm$ s.e.) | I ($\pm$ s.e.) | $R^2$ |
|---|---|---|---|---|---|
| $\delta^{18}O_{obs} \sim \delta18O_{mod} + I$ | 4503 | 0.761 | 0.917 ($\pm$ 0.012)* | -0.645 ($\pm$ 0.168)* | 57.9% |
| $\delta^{18}O_{obs,avg} \sim \delta^{18}O_{mod,avg} + I$ | 448 | 0.820 | 0.879 ($\pm$ 0.029)* | -0.891 ($\pm$ 0.414)* | 67.3% |
| $\delta^2 H_{obs} \sim \delta^2 H_{mod} + I$ | 4503 | 0.819 | 0.937 ($\pm$ 0.010)* | -1.90 ($\pm$ 1.06)* | 67.1% |
| $\delta^2 H_{obs,avg} \sim \delta^2 H_{mod,avg} + I$ | 448 | 0.866 | 0.905 ($\pm$ 0.025)* | -3.10 ($\pm$ 2.66) | 75.1% |
| $\delta^2 H_{diff} \sim \delta^{18}O_{diff} + I$ | 4503 | 0.959 | 6.54 ($\pm$ 0.029)* | 1.30 ($\pm$ 0.065)* | 91.9% |
| $\delta^2 H_{avg,diff} \sim \delta^{18}O_{avg,diff} + I$ | 448 | 0.958 | 6.70 ($\pm$ 0.094)* | 1.46 ($\pm$ 0.190)* | 91.9% |

**Table 1.** Correlation and regression results for observation-model comparisons. Regressions were performed on all data (n = 4503), as well as on the mean values in a subset of the reaches with more than one observation (n=448). An asterisk (*) indicates the coefficient is significant at p<0.1.

the modeling framework, high elevation groundwater discharge, does not bear an evapoconcentrated isotopic signature in our input datasets, and lower elevation water sources (groundwater or runoff) that could bear a signature of evapoconcentration, depending on the region, are considered by the model to be minor contributors to streamflow over the timescale integrated by our study.

### 3.2 Model-observation differences

Of 4503 observations, 1763 $\delta^{18}O$ and 3306 $\delta^2 H$ observations were significantly different from the long term mean isotope mass balance NWM estimate at p<0.1. Of these, 1756 observations indicated significant differences in both isotopologues. This corresponded to a median absolute difference of 2.2‰ for $\delta^{18}O$ and 9.7‰ for $\delta^2 H$. For both, a larger proportion of the distribution indicated positive significant differences and those differences tended to be greater in absolute magnitude than the negative significant differences.

We used an observation-model difference interpretation framework (Figure 2) to interpret differences between the observations and the isotope mass balance estimates. The differences may contain process information that can be used to improve our understanding of terrestrial water balance and process inclusion in the NWM. The observation-model differences in $\delta^{18}O$ and $\delta^2 H$ were correlated (Figure 4) and yielded similar results for analyses performed with all data as compared to means of reaches with multiple observations (Table 1). Correlations between $\delta^{18}O_{diff}$ and $\delta^2 H_{diff}$ were about 0.96. Simple linear

regressions, where variance in $\delta^{18}O_{diff}$ explained variance in $\delta^2 H_{diff}$, with all data and catchment mean data both explained about 92% of the variance, were significant and exhibited slopes of less than 8 (Table 1), suggesting the presence of errors arising from NWM omission of water sources that bear signatures of non-equilibrium processes.

    In our dataset, model estimates do not deviate much from the GMWL, and deviate less than the observations (Figure 3). This information is quantified using $d_{diff}$ (Figure 2). Positive values of $\delta^{18}O_{diff}$ tended to be associated with negative values of

$d_{diff}$ (Figure S8). The shape of the relationship between the two quantities is non-linear, with a stronger relationship between $\delta^{18}O_{diff}$ and $d_{diff}$ among data from arid reaches compared to humid reaches.

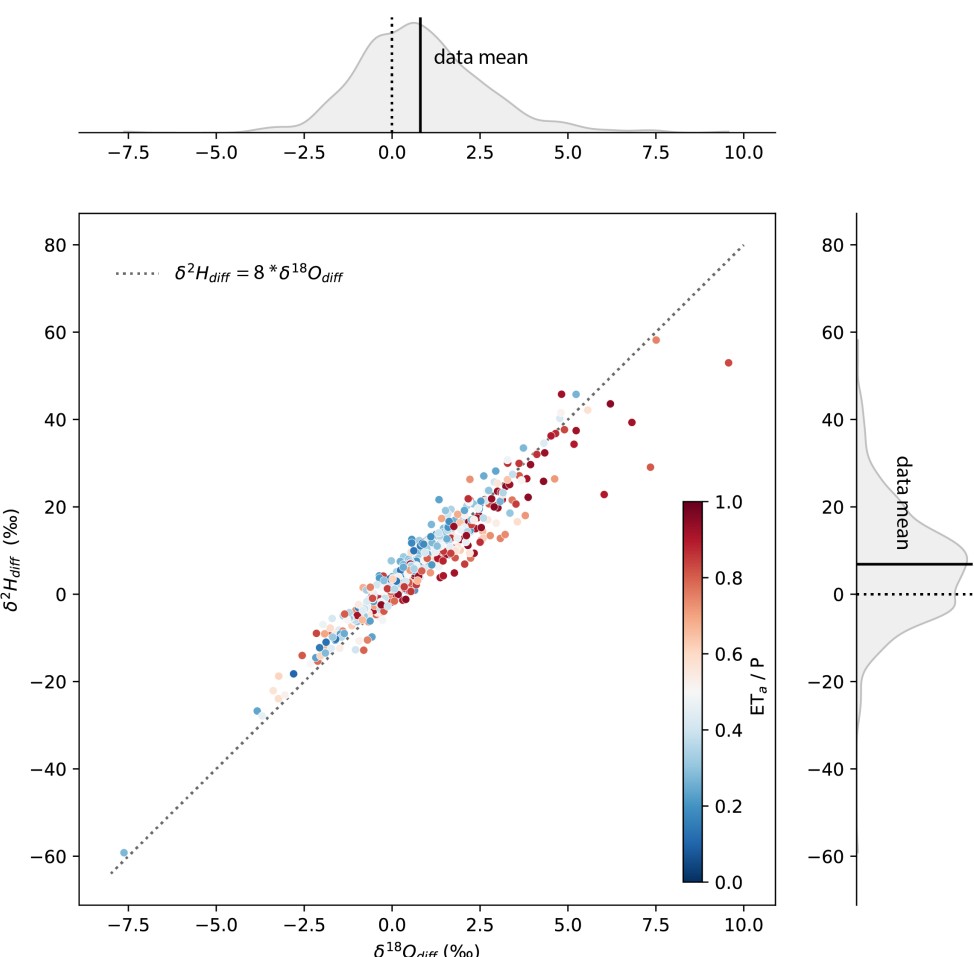

**Figure 4.** The relationship of observation (obs) - isotope mass balance (mod) estimation differences for $\delta^{18}O$ and $\delta^2H$. The catchment mean value is plotted, and only sites with at least two observations are shown (n=448). The equilibrium line with slope 8 is plotted for context (dotted line), and data are colored by their site's the ratio of actual evaporation to precipitation. Data distributions are shown for both $\delta^{18}O_{diff}$ and $\delta^2H_{diff}$ in the margins, along with the mean differences indicated as a solid line. No difference (0) is marked with a dotted line for reference.





| Surface water lines: | $\beta$ ($\pm$ s.e.) | I ($\pm$ s.e.) |
|---|---|---|
| Model-derived | 8.12 ($\pm$ 0.010) | 8.06 ($\pm$ 0.14) |
| Observations | 7.57 ($\pm$ 0.02) | 1.23 ($\pm$0.32) |
| Meteoric water lines: | $\beta_{min}$, $\beta_{max}$ ($\beta_{avg}$) | $I_{min}$, $I_{max}$ ($I_{avg}$) |
| Global Meteoric Water Line | 8 | 10 |
| Arid and Temperate dry summer LMWLs | 6.56, 8.02 (7.57) | -10.5, 9.85 (3.02) |
| Temperate humid and Continental LMWLs | 7.34, 7.64 (7.49) | -3.82, 3.31 (0.62) |

**Table 2.** Surface water line slopes and intercepts ($\delta^2 H = \delta^{18}O + I$) compared to the Global Meteoric Water Line and published precipitation water line ranges (LMWLs) from different climate classifications in North America (data from Putman et al. (2019)).

The relationship between $\delta^{18}O_{diff}$ and $d_{diff}$, as well as our regression (Table 1) and surface water line analyses (Table 2) indicate that the modeling approach for estimating long term isotope ratios of rivers produce results that are similar to, but on average, lower and exhibit less variability than observations. The strongest signal in our data is that of evaporation, evi-
denced by combinations of positive $\delta^{18}O_{diff}$ and negative $d_{diff}$ in arid regions. We also observe evidence of non-equilibrium condensation processes in reaches characterized by negative $\delta^{18}O_{diff}$ and positive $d_{diff}$.

We suggest that patterns in $\delta^{18}O_{diff}$ and $d_{diff}$ contain useful model diagnostic information that can be useful for improving the NWM and our understanding of the terrestrial water balance. The modes of variability we evaluate include 1) interannual 2) monthly and 3) spatial. The first is to evaluate the effect of comparing long term mean mass balance estimates with sample
collection. The second and third modes of variability provide information about the presence of possible missing water sources to the model. Additional sources of variability are discussed in Text S6.

### 3.3 Interannual variability in observation-model differences

The annual mean $\delta^{18}O_{diff}$ and $d_{diff}$ values for NRSA collection years were statistically indistinguishable from the mean differences calculated with the whole dataset (Figure 5). Though all observation-model difference regression slopes were
less than 8, and all intercepts were greater than 0, the observation-model difference regression slopes and intercepts were significantly different from one another and the regression calculated from all available data.

We interpret the year to year similarity in the regression slopes and intercepts as indicating the pervasive presence of evapoconcentrated observations in the region. We interpret the variability in the slopes and intercepts as arising from interannual differences in climate. Such interannual differences may include variability in snowpack isotope ratios and the proportion of
runoff from rain or snowmelt in the stream relative to groundwater. We suggest that the year to year variability in slopes and intercepts is more likely to derive from climatic differences than differences in sampling distribution.

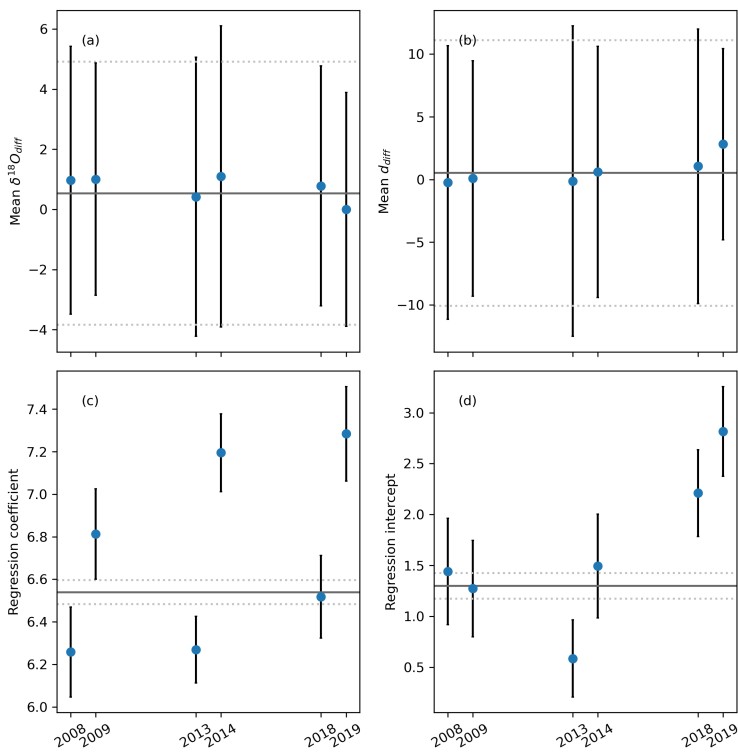

**Figure 5.** Interannual variability in observation-model comparisons using data from years with at least 250 samples in 100 different reaches (NRSA years). (a) The annual mean (blue points) and 95% confidence intervals (solid vertical lines) of $\delta^{18}O_{diff}$ compared to the mean (solid horizontal line) and 95% confidence intervals (dotted horizontal lines) of all data. (b) The annual mean and 95% confidence intervals of $d_{diff}$ compared to the mean and 95% confidence intervals of all data. (c) The slope of $\delta^2 H_{diff} \sim \delta^{18}O_{diff}$ and 95% confidence intervals compared to the slope and 95% confidence intervals of calculated from all data. (d) The regression intercept of $\delta^2 H_{diff} \sim \delta^{18}O_{diff}$ and 95% confidence intervals compared to the regression intercept and 95% confidence intervals of calculated from all data. In all subplots, the all-data mean and 95% confidence interval are plotted for reference



### 3.4 Seasonal variability in observation-model differences

There are systematic patterns in $\delta^{18}O_{diff}$ and $d_{diff}$ when examined across the growing season. For example, $\delta^{18}O_{diff}$ tends to be greater during the latter months of the growing season relative to the mean $\delta^{18}O_{diff}$ value for the month of June for

that site and year (Figure 6 a) in most basins and months. The pattern is especially evident in the Great Basin and California. Likewise, $d_{diff}$, is lower in July, August, September relative to June (Figure 6 b), but only in the Great Basin and California. The contrast between basins with both increased $\delta^{18}O_{diff}$ and decreased $d_{diff}$ (Great Basin and California) and those with only increased $\delta^{18}O_{diff}$ and little change in $d_{diff}$ (Upper and Lower Colorado and Pacific Northwest) suggests that two different mechanisms may drive isotopic change during the growing season.

In California and the Great Basin, which are characterized by larger [18]O-enrichment and $d$ decrease over the growing season, we suggest increased contributions of evapoconcentrated waters to rivers later in the growing season. In California, this may reflect the water use in the Central Valley.

In the Upper and Lower Colorado, and Pacific Northwest, where we observe small [18]O enrichment in the absence of a notable change in $d$ the second case, we suggest sustained dependence on the same water source regions throughout the growing season.

In the Pacific Northwest and parts of the Upper Colorado, the relative invariability may indicate the sustained dependence on groundwater discharge from high elevations to streamflow during the growing season (Miller et al., 2016; McGill et al., 2021; Windler et al., 2021). In lower parts of the Upper Colorado and the Lower Colorado, where rivers are characterized by large reservoirs, the seasonal invariance may reflect that the primary 'water source' regions for these reaches are reservoirs, which retain snowmelt from early in the season and discharge it later in the season. In this way, the primary 'water source' for the

streams in many areas of these basins may remain constant throughout the growing season.





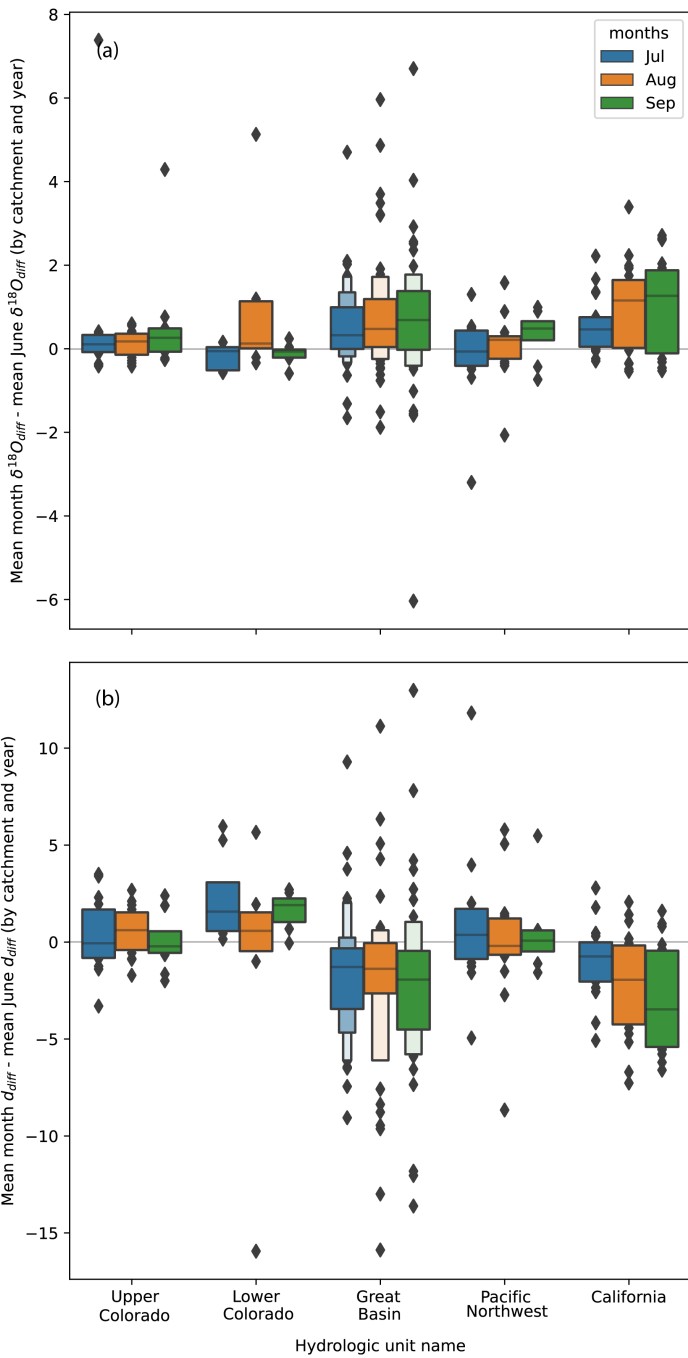

**Figure 6.** Evaluation of seasonal variability in observation-model comparisons. Data include all reaches and years with collections in the month of June as well as two of the 3 other months of the summer season. (a) The distribution of month-specific differences from June $\delta^{18}O_{diff}$ by basin. (b) The distribution of month-specific differences from June $d_{diff}$.



## 3.5 Spatial distribution of observation - model differences

If the NWM fully constrained all relevant water sources, we expect to observe similar values of $\delta^{18}O_{diff}$ and $d_{diff}$ throughout each basin, irrespective of the location of the observation in the basin. Instead, we observed spatial variability (Figures 7 and S9), where smaller magnitude $\delta^{18}O_{diff}$ values occurred in the highest elevation, lowest stream order, and least arid reaches, and larger magnitude, often positive $\delta^{18}O_{diff}$, values occurred in lower elevation, arid or intermittent flow reaches (Figure S10). $d_{diff}$ tended to exhibit higher values in higher elevation, lower stream order reaches, and lower values in lower elevation, more arid, higher stream order reaches (Figure 8). We observed a greater range in the absolute magnitudes of $\delta^{18}O_{diff}$ and $d_{diff}$ in higher order, lower elevation reaches (Figures 8 and S10). Notably, the pattern was similar across basins, suggesting the importance of within-basin processes in determining $\delta 18 O_{diff}$ an d$d_{diff}$, as opposed to absolute relationships of $\delta^{18}O_{diff}$ and $d_{diff}$ to elevation, stream order, or climate classification.

The spatial pattern in $d_{diff}$ (Figure 7) was similar to the pattern observed for the KGE and other metric evaluations of the NWM (Towler et al., 2023). Areas with negative $d_{diff}$ tended to correspond to areas with poor NWM performance (Towler et al., 2023). However, the isotopic evaluation of NWM and the Towler et al. (2023) datasets could not be directly compared due to a there being only a small number of reaches with both isotope observations and daily discharge measurements.

The spatial structure of our results was statistically well explained by the the ratio of actual evaporation to precipitation ($\frac{ET_a}{P}$) in a linear mixed effects model with basin as the grouping variable (Table 3). Variability among basins explained 16.2% of the variance in $d_{diff}$, while the fixed effect of aridity explained 13.9% of the variability in the dataset. The regression slope associated with the fixed effects of aridity was negative (-7.87$\pm$ 0.78) and significant (p<0.001), indicating that sites with higher aridity indices tended to exhibit more negative $d_{diff}$. This regression was stronger than a linear mixed effects model with elevation predicting $d_{diff}$, where the fixed effects of elevation explained 4.7% of the variability in $d_{diff}$.

Analysis of the spatial variability in our results suggest that 1) higher elevation, lower stream order, perennial, warm temperate or seasonally snowy reaches had small $\delta^{18}O_{diff}$ and positive $d_{diff}$ values and 2) lower elevation, higher stream order, arid and sometimes intermittant stream reaches had larger and more positive $\delta^{18}O_{diff}$ values and more negative $d_{diff}$ values. The first point suggests errors associated with the challenges of providing input values at appropriate temporal resolutions, whereas the second point suggests the model is missing critical evapoconcentrated water sources in more arid, lower elevation areas of each basin.



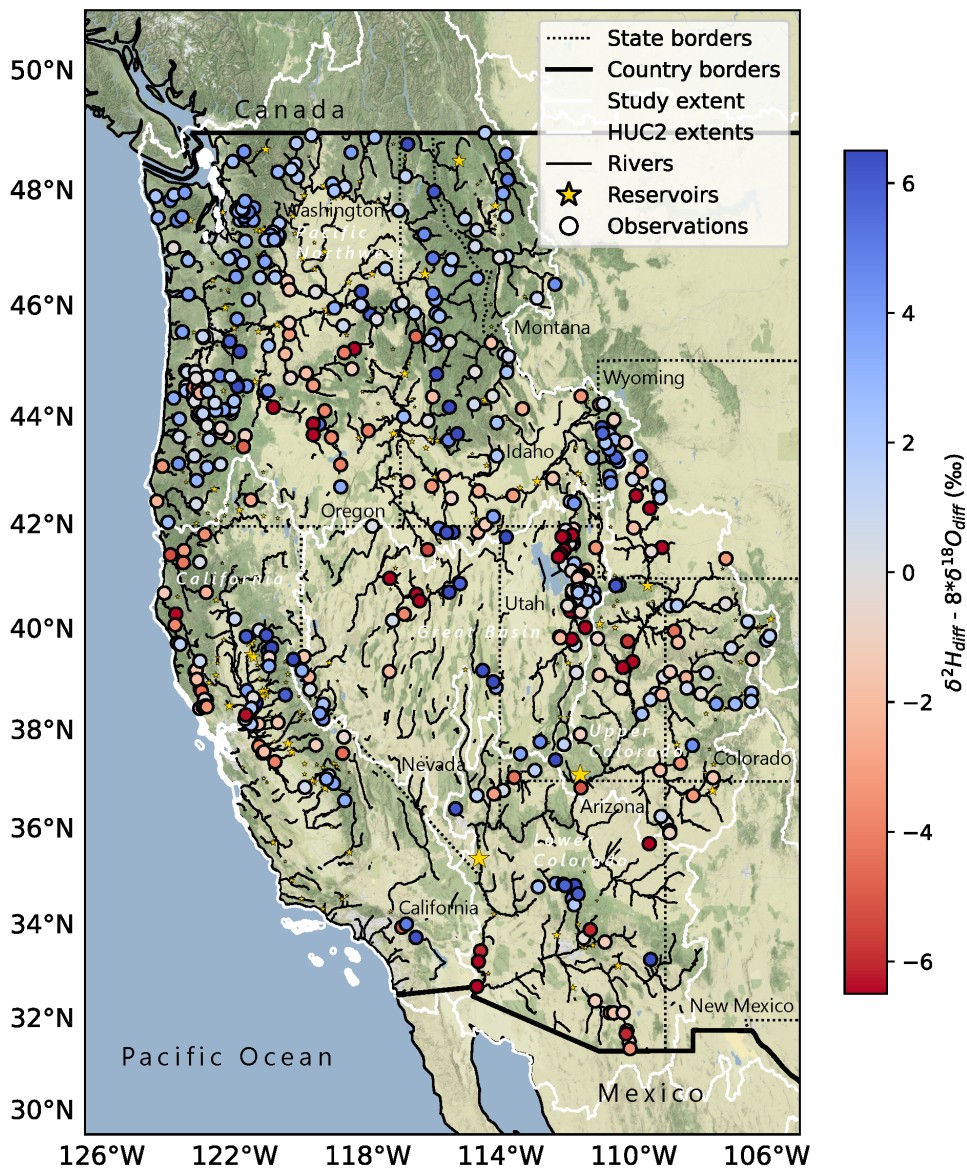

**Figure 7.** The spatial distribution of mean catchment $d_{diff}$ ($\delta^2 H_{diff} - 8 * \delta^{18}O_{diff}$) in reaches with more than one observation (n=448). Locations of reservoirs are marked by yellow stars, with the star size proportional to the reservoir capacity. Redder colors indicate more evapoconcentrated waters than expected based on the model estimate. Map data is from ©OpenStreetMap contributors 2023, distributed under the Open Data Commons Open Database License (ODbL) v1.0, accessed through Stamen OpenSource Tools (https://stamen.com/open-source/). HUC2 basins come from WBD (U.S. Geological Survey, National Geospatial Technical Operations Center, 2023) and rivers are modified from the NHDPlus streamline network (U.S. Geological Survey, 2019).



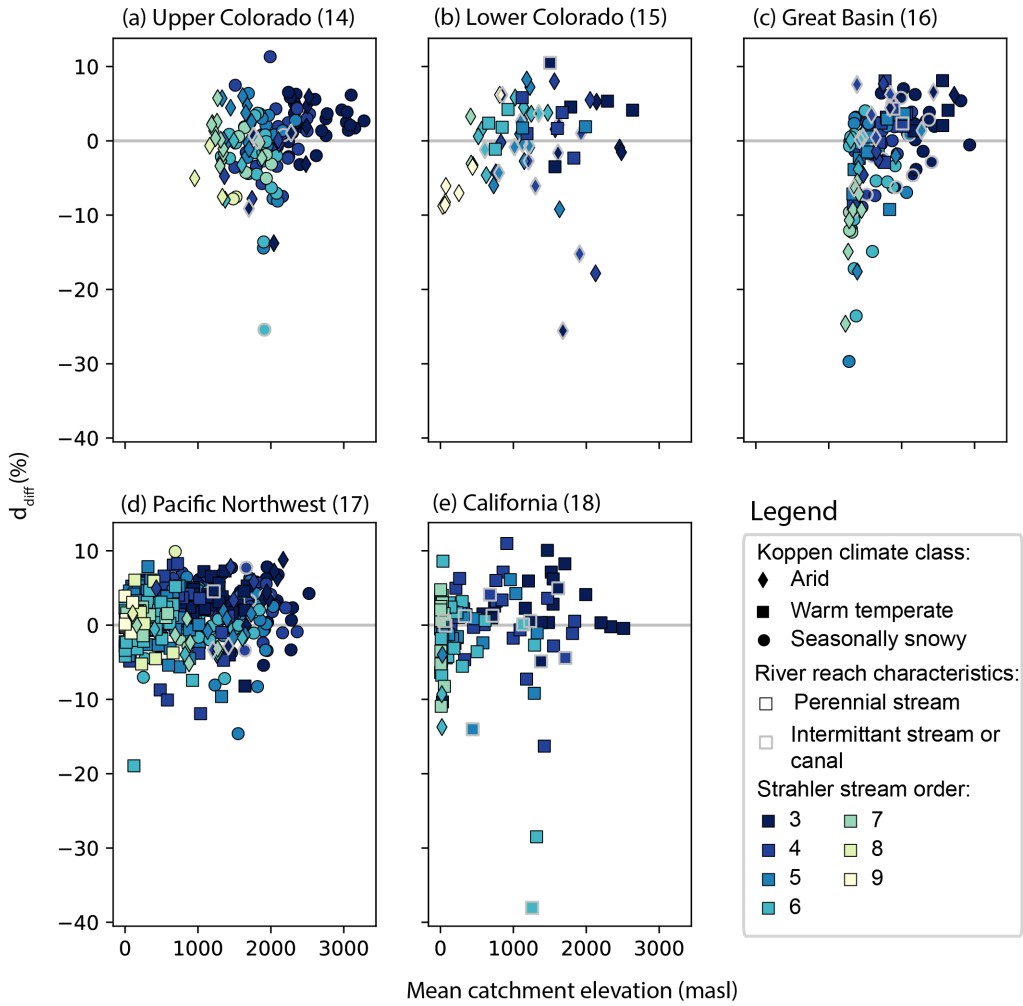

**Figure 8.** Relationship of elevation, Strahler stream order, and Köppen climate classification (Rubel and Kottek, 2010), and stream persistence to $d_{diff}$ in each basin. We observe higher $d_{diff}$ in perennial, lower order streams at middle and higher elevations in each basin. Lower $d_{diff}$ is associated with higher order streams at lower elevations in each basin. This effect was greater in catchments classified as arid or seasonally snowy compared to those classified as warm temperate. This pattern was generally true in each basin, irrespective of the absolute elevation or stream order, suggesting the importance of accumulated effects within a basin on $d_{diff}$.



### 3.5.1 Observation-model differences in headwater reaches reflect groundwater isotope ratio estimates

We observe statistically significant reach-scale $\delta^{18}O_{diff}$ and $d_{diff}$ in higher elevation, low stream order, low aridity, temperate or seasonally snowy reaches in our dataset (Figures 8, S10). These differences tend to be smaller than than the full dataset mean
$\delta^{18}O_{diff}$ and $d_{diff}$. At most of these reaches we also observe positive $d_{diff}$ values (Figures 7, 8).

The presence of both negative and positive values of $\delta^{18}O_{diff}$ likely reflect interannual variability in the isotope ratios of actual groundwater and snowmelt discharged to rivers in high elevation headwater areas. Although groundwater's contribution to streams is conceptualized in this study to be constant in magnitude and isotope ratio, in reality, the isotope ratios of both groundwater and snowmelt fluxes vary spatially and interannually. The groundwater flux magnitudes vary interannually based
on variations in snowpack magnitudes, antecedent hydrologic conditions,  (Brooks et al., 2021; Wolf et al., 2023) and hydrogeologic (Gentile et al., 2023) controls including hydrologic residence times. Snowpack isotope ratios vary in response to climate patterns and local conditions (Anderson et al., 2016). The observed variability of $\delta^{18}O_{diff}$, which does not exhibit a uniform tendency towards positive or negative values, suggests the mean groundwater isotope ratios used in this study are reasonably representative of the long term mean estimates of the isotope ratios of water contributed at high elevation water source areas by
groundwater and snowmelt fluxes. Though improvements may be made by using interannually varying estimates of the isotope ratios of groundwater and snowmelt. However, the systematic positive $d_{diff}$ result cannot be explained by the timescale of the isotope input.

Higher $d$ streamflow relative to weighted mean precipitation values have been documented in other studies (Nickolas et al., 2017). This may be because higher $d$ is associated with lower precipitation $\delta^{18}O$ that falls during the cold season in mid-
latitude regions, particularly in areas near open water (Putman et al., 2019; Corcoran et al., 2019; Aemisegger and Sjolte, 2018). Secondarily, high $d$ in rivers relative to precipitation or groundwater may be attributed to fractionation occuring during melt. The snow melt process has been demonstrated to begin with preferential melt of water molecules bearing lighter isotopologues, and to exhibit higher $d$ earlier in the melt season (Ala-aho et al., 2017; Beria et al., 2018; Carroll et al., 2022). The higher $d$ of the snow and initial meltwater may be passed along to the rivers via direct runoff to streams or through shallow groundwater
recharge and rapid discharge to streams (see the relatively higher upper bound on $d$ values for forested land use types in Figure 9).




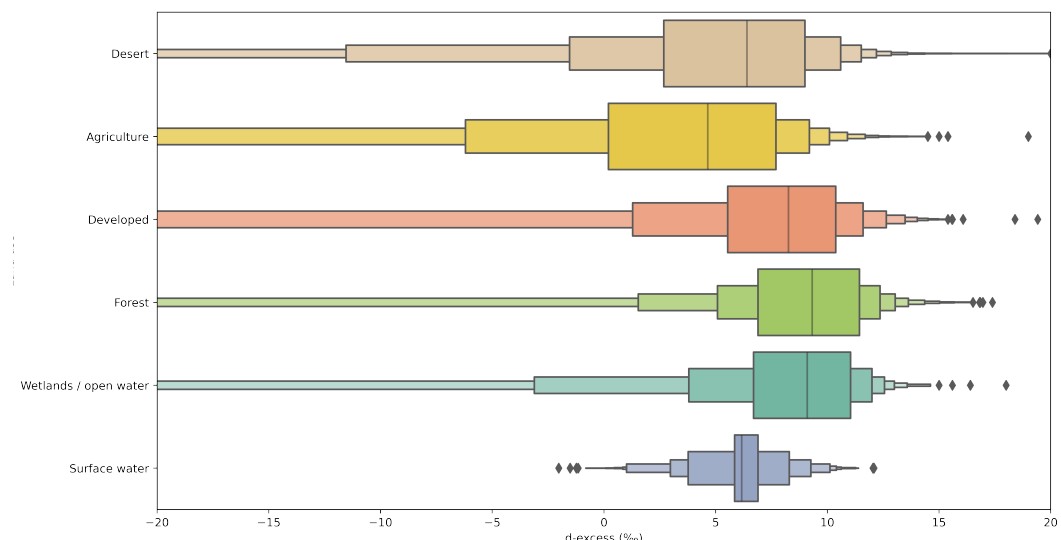

**Figure 9.** Distributions of groundwater $d$ grouped by the dominant land type from NLCD (Dewitz and U.S. Geological Survey, 2021) in the HUC12 (U.S. Geological Survey, National Geospatial Technical Operations Center, 2023) of the observation. The desert land class includes barren land (often playas or dried lakebeds), shrub/scrub, grasslands/herbaceous. The agricultural land class includes pasture/hay and cultivated crops. The developed land class includes developed land of any intensity. Forest includes evergreen, deciduous and mixed forest. The wetlands/open water land class category any type of wetland as well as open water. The distribution of our 4303 river samples is also shown for context.

### 3.5.2 Evapoconcentration at low elevations suggests contributions of irrigation return flows to streamflow

Greater spatial and temporal variability in both $\delta^{18}O_{diff}$ and $d_{diff}$ in lower elevation, higher stream order, arid reaches suggests the importance of various spatially and temporally heterogeneous processes and water sources that may alter streamflow isotope ratios relative to upstream values. Positive values of $\delta^{18}O_{diff}$ and negative values of $d_{diff}$ in more arid regions of each basin suggests that evapoconcentrated waters compose a large fraction of streamflow in these areas (Figures 7, 8, S9 and S10), especially in the later part of the growing season (Figure 6) when streams depend more heavily on groundwater fluxes. We observed evidence of surface water evapoconcentration in all basins (Figure 8), though it was most apparent in Lower Colorado River Basin, lower elevation regions of the Upper Colorado River Basin, California's Central Valley, near Great Salt Lake in the Great Basin and throughout the Snake River Plain (Figures 7 and S9).

The isotope ratios and $d$ we observe in low elevation, high stream order arid reaches are similar to those we would expect to observe in highly evaporative contexts, like within lakes (Bowen et al., 2018), intermittent flow rivers, or downstream of wetlands. Yet the majority of rivers in our study are perennial, and most are not characterized by substantial wetlands.





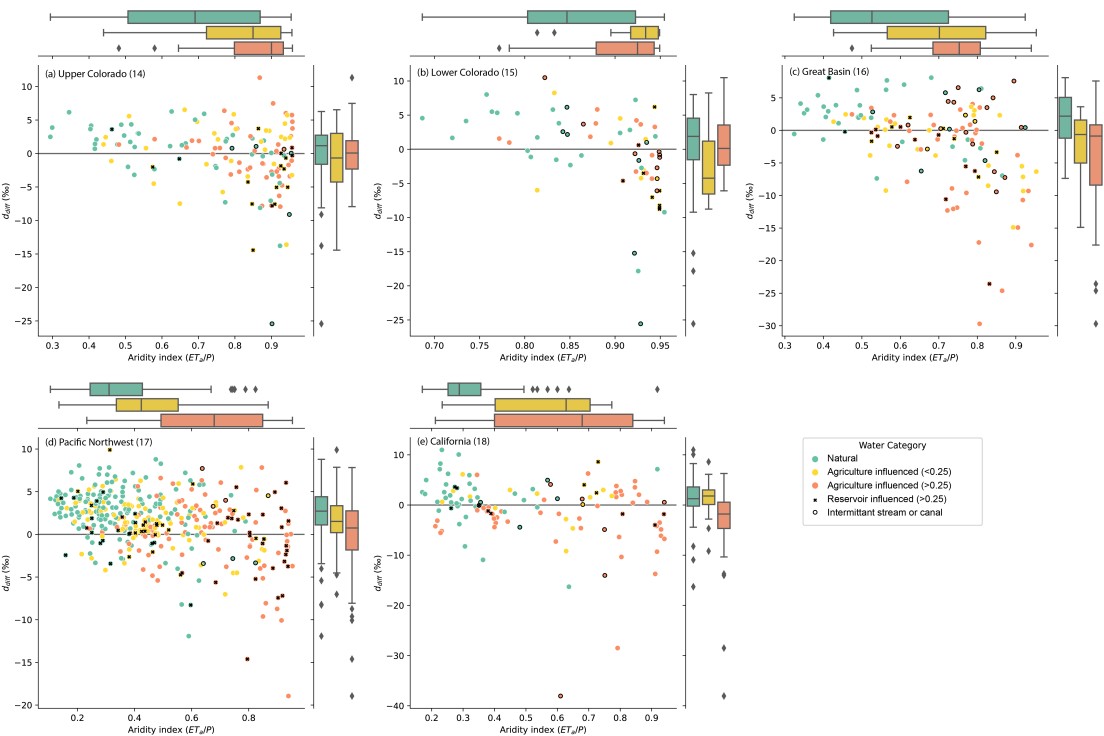

**Figure 10.** Relationship of aridity to $d_{diff}$, by water use categories and basins. Natural waters are not estimated to be influenced by agricultural irrigation. Fractions of agricultural irrigation contributing to streamflow are estimated using water use data and land cover data and do not account for losses to evapotranspiration. Reaches affected by large reservoirs and reaches categorized as intermittent or as canals or ditches are indicated.

The evapoconcentration in our dataset is unlikely to arise from river or reservoir evaporation because both evaporation of reservoirs and evaporation to inflow ratios in the region tend to be low, especially for deep man-made reservoirs (Brooks et al., 2014; Friedrich et al., 2018). Instead, isotopic evidence of evapoconcentration occurs in waterways likely to be affected by anthropogenic hydrologic alteration (Fergus et al., 2021) and characterized by larger fractions of 'young water' (Jasechko et al., 2014; Burt et al., 2023; Xia et al., 2023).

We tested the hypothesis that the spatial pattern in isotopically-inferred evapoconcentration could arise from contributions of irrigation return flows to streams and reservoir releases. Within each basin, on mean, $d_{diff}$ was most negative at sites with the highest proportion of total inflows attributed to agricultural return flows and highest at sites with no apparent contributions of agricultural return flows (Figure 10). Reservoir influence was associated with low $d_{diff}$ more often where dams are used for water management and water supply (e.g., Upper Colorado, Lower Colorado, Great Basin, and California) and were associated with high $d_{diff}$ in the Pacific Northwest, where dams are more often used for hydropower. Intermittent streams and canals in arid regions were sometimes associated with low $d_{diff}$ as well, even when no water was contributed by agricultural irrigation.





We demonstrated the relationships of agricultural and reservoir influence on $d_{diff}$ statistically in a linear mixed effects model (Table 3). The fraction of streamflow estimated to come from agricultural irrigation return flows and a categorical variable delineating reservoir influence together explained 7.7% of the variance in $d_{diff}$, with the whole model (including random group effects) explaining 14.5% of the variance in the dataset. Both explanatory variables were significant (p<0.001), and, as expected, exhibited negative slopes indicating that greater agriculture and reservoir influences tended to produce lower, more evapoconcentrated $d_{diff}$ values. When we also include the ratio of actual evaporation to precipitation with these explanatory variables, all three are significant (p<0.01) and explain 15.1% of the variance through fixed effects, and 23.4% of the variance overall (fixed and random effects). Among the linear mixed effects models tested, it exhibited the highest log likelihood value, explained the greatest amount of variance using fixed effects, and reduced the amount of variance attributed to random within-basin effects.

While this statistical model performance is not substantially better at explaining variance in $d_{diff}$ than the model that uses aridity alone, the findings do suggest that both agricultural activity and reservoirs influence the isotope ratios of streamflows across the Western US. The low variance explained by these models is expected, due to the difficulty estimating a true long term mean agricultural return fluxes with the available data and the potential for isotopically heterogeneous reservoir effects, as well as the covariance of both irrigation return flows and the presence of reservoirs with aridity and elevation. However, our findings are well supported by studies of irrigation-based contributions to streamflow, an analysis of groundwater evapoconcentration by land use, comparison of the Jasechko et al. (2021) dataset with NWM groundwater contributions to streams in the context of patterns in agricultural irrigation.





| Statistical model | $\beta$ ($\pm$ s.e.) | I ($\pm$ s.e.) | HUC2 Var | Cond. $R^2$ | Fixed $R^2$ | Log likeli-hood |
|---|---|---|---|---|---|---|
| $d_{diff} \sim Elev + I$ | $Elev$: 0.001 (0.00)* | -1.93 (1.01) | 4.33 | 20.9% | 4.4% | -2254 |
| $d_{diff} \sim \frac{ET}{P} + I$ | $\frac{ET}{P}$: -7.85 (0.77)* | 4.86 (1.08)* | 4.37 | 30.2% | 13.9% | -2209 |
| $d_{diff} \sim F_{irr} + Res + I$ | $F_{irr}$: -3.49 (0.49)*  $Res$: -1.76 (0.45)* | 0.92 (0.59) | 1.464 | 14.3% | 7.9% | -2225 |
| $d_{diff} \sim \frac{ET}{P} + F_{irr} + I$ | $\frac{ET}{P}$: -6.53 (0.83)*  $F_{irr}$: -1.61 (0.54)* | 4.40 (0.35)* | 1.942 | 22.7% | 14.7% | -2203 |
| $d_{diff} \sim \frac{ET}{P} + F_{irr} + +Res + I$ | $\frac{ET}{P}$: -6.11 (0.88)*  $F_{irr}$: -1.68 (0.54)*  $Res$: -1.22 (0.44)* | 4.33 (0.82)* | 1.861 | 23.0% | 15.2% | -2199 |

**Table 3.** Results of linear mixed effects models with 764 observations and 5 groups. The minimum and maximum group sizes were 48 and 387, respectively. The models do not include any samples from reaches characterized as an intermittent stream or canal or where NWM indicates that the maximum streamflow is 0 m$^3$ s$^{-1}$. Random effects apply only to the intercepts. An asterisk indicates that a regression coefficient is statistically significant at p<0.01. Conditional $R^2$, which gives the total model variance explained, are reported alongside the fixed $R^2$, which gives the variance explained by fixed effects (i.e., explanatory variables) and the log-likelihood, which can be used to evaluate the relative performance of different models.

### 3.5.3 Literature and datasets support isotopic inference of irrigation return flows contributing to streamflow

Numerous prior studies have investigated the influence of irrigation on streamflow. Estimates suggest that, depending on the irrigation type, as much as 50% of applied water may recharge groundwater or arrive at surface waters through shallow groundwater infiltration and subsequent discharge to streams (Grafton et al., 2018). Likewise, irrigation has been demonstrated to increase streamflows during low flow periods (Fillo et al., 2021; Essaid and Caldwell, 2017), if the applied water comes from surface water diversions.

Local contributions of groundwater to streams from irrigation-based recharge are supported by the $d$ values of groundwater in agricultural regions. Groundwater from regions influenced by agricultural irrigation exhibited lower mean $d$ relative to deserts, including dried terminal lakes and playas, developed areas which may include turf grass irrigation, forested regions, wetlands or open waters and surface waters (Figure 9). Some contribution of this irrigation-recharged groundwater to streams via return flows would decrease the difference between the modeled and observed isotope ratios in our dataset, and is supported

by conclusions from prior isotopic inference of water sources in the Snake River plain (Windler et al., 2021).

The isotopic inference that irrigation return flows are an important missing process in the NWM is supported by an independent statistical comparison of the NWM groundwater discharge with the Jasechko et al. (2021) well water level comparison to stream level dataset and the agricultural water use data. We hypothesize that if NWM accurately represents groundwater discharge to streams, the Jasechko et al. (2021) well water level comparison to stream water level dataset should be able to pre-

dict the summer mean NWM groundwater discharge flux with a large proportion of variance explained. However, the Jasechko





et al. (2021) data (expressed as the fraction of well water levels that lie below the proximal river water level) weakly, though significantly ($R^2$ = 0.028, p<0.001) predict the NWM groundwater discharge rates in a simple linear regression. The regression relationship between the variables is negative, as expected, where river reaches with a greater proportion of their well water levels above proximal river water levels correspond to reaches with greater groundwater discharge fluxes (Figure S11). Though

the regression is significant, it has almost no predictive capacity, contrary to what we expect if the well water level comparison to steam level dataset was a good predictor of the NWM groundwater discharge to streams.

     The weakness of the statistical relationship between the well water level comparison to river water level and the NWM groundwater discharge flux may be related to shallow aquifers, which are not considered by NWM, and/or agricultural irrigation, as well as the water source (surface or ground water) used for that irrigation (Figure S12). We did not assess the potential

for NWM groundwater discharge to reflect the presence of shallow aquifers. However, we do observe that the influence of irrigation on groundwater levels is non-stationary, depending on both the groundwater discharge level as well as the source of irrigation water. For this reason the relationship is difficult to assess statistically. In river reaches where NWM indicates little groundwater discharge ($0^{th}$ to $20^{th}$ percentile qBucket), irrigation sourced from surface water is associated with a larger fraction of well water levels above river water level in a catchment than those without irrigation. Conversely, in river reaches with

substantial groundwater discharge ($80^{th}$ to $100^{th}$ percentile qBucket), agricultural irrigation with water from either surface or groundwater tends to be associated with a smaller fraction of well water levels above river level in the catchment compared to reaches without any agricultural irrigation. Based on these patterns we suggest that in dry areas, irrigation from surface water appears to contribute to groundwater recharge, whereas in wet areas, irrigation appears to contribute to decreased water table elevations. Surface water irrigation tends to contribute to higher water tables, whereas irrigation from groundwater tends to

contribute to lower water tables.

     Some part of this signal is regional. Reaches from more arid basins compose a greater proportion of the lower percentile qBucket reaches, and reaches from humid or seasonally snowy basins compose a greater proportion of the higher percentile qBucket reaches. However, when evaluated by basin, the relationships are similar. The finding is consistent with modeling studies, which showed lower stream discharge when irrigation water came from groundwater, and greater stream discharge

when irrigation water came from surface water (Essaid and Caldwell, 2017). Our analysis suggests that agricultural irrigation is likely to influence groundwater levels and groundwater discharge on a landscape scale and produces gaining streams and contributes to streamflow in otherwise arid, losing reaches of rivers.

### 3.6   Implications of including irrigation return flows into NWM calculations

Our evaluation of the NWM-driven isotope mass balance calculations suggest that the NWM accuracy would be improved

by including agricultural return flows in the water sources sustaining streamflow in the NWM. In effect, agricultural return flows are simply groundwater fluxes to streams that occur at lower elevations than the majority of the groundwater discharge sustaining streams. Based on magnitudes of $d_{diff}$, these lower elevation groundwater fluxes can sometimes be large. Because the NWM is calibrated to actual streamflows which contain these return flows, these fluxes are currently being misallocated in the model. Inaccuracies in any model terms or fluxes influence the model's capacity to project accurate streamflows, particularly





under non-stationary hydrologic conditions. Accurate model water source inclusion, particularly at low elevations where water use and availability is most critical, thus has implications for the model's utility to stakeholders, including water managers and users.

Under current conditions, agricultural return flows may be critical for sustaining streamflow late in the growing season (August or September) or during drought periods. Sustained streamflow in certain reaches is critical for 1) water access for surface

water diversions and 2) water availability for species' use. For example, protected fish species survival requires that waterways meet thresholds of water quality, temperature, and depth for survival (Dibble et al., 2020). Water managers make decisions about water allocations and reservoir releases in part to meet these habitat needs (Bruckerhoff et al., 2022). Agricultural return flows have the capacity to help sustain streamflow (Fillo et al., 2021), but with potentially negative effects on water quality, through agriculture-associated salinization (Miller et al., 2017; Thorslund et al., 2021), increased concentrations of nitrate (Lin

et al., 2021), and other nutrients (Stets et al., 2020), contributions of pesticide and fertilizers, or alterations to water temperature profiles. These contributions of agricultural waters contribute to sustaining flow but threaten water availability. Thus, inclusion of groundwater return flows from irrigation to rivers in the Western US supports improved assessments of water availability both through improved modeling of streamflows and enhanced ability to model water quality.

Explicit inclusion of irrigation return flows will assist the NWM in better projecting streamflows during periods of hydrologic

non-stationarity, as are likely to characterize the hydroclimatic elements of climate change. Non-stationary processes include hydrologic changes arising from the ongoing mega-drought of the southwestern US (Williams et al., 2022), intense precipitation events like monsoons or major storm events that are observed to be increasing in intensity with climate change (Pfahl et al., 2017; Demaria et al., 2019), and projected changes to future snowpack depth and melt timing (Siirila-Woodburn et al., 2021; Hammond et al., 2023). The ongoing aridification of the southwestern US is characterized by increased evapotranspira-

tion (Milly and Dunne, 2020), and changes to groundwater recharge and discharge associated with decreases in snowpack and changes to snowpack melt patterns (Hammond et al., 2023). Understanding the groundwater flux contributions of areas with shallow water tables to streamflow during major precipitation events will help better characterize areas at risk for flooding and inform appropriate water management strategies.

## 4   Conclusions

The isotope mass balance evaluation of the NWM revealed similarities between the isotope mass balance estimated isotope ratios (modeled) and observed isotope ratios. The mass balance approach captured as much as 75% of the variance in the observations, depending on the water isotopologue evaluated. This suggests that, on mean, during the summer, the NWM correctly represents the relative proportions of groundwater and runoff fluxes sustaining streamflow, and the gridded isotope datasets are appropriate for the analysis.

The observation-model differences exhibited spatial and seasonal structure, suggesting that the NWM is missing important additional water sources that contribute to streamflow. Specifically, the observation-model differences that plot above the equilibrium line (Figure 2) suggest the importance of direct contributions of snowmelt to streamflow in humid areas. Those that

plot below the equilibrium line suggest the importance of groundwater sources characterized by evapoconcention in arid areas. We tested the hypothesis that agricultural irrigation return flows are the missing, isotopically evapoconcentrated water source

in arid regions, and found them to be a significant predictor of observation-model differences. This finding is supported by multiple lines of evidence including land use pattern influence on isotopic evapoconcentration of groundwaters, a comparison of NWM groundwater discharge and an independent assessment of the potential for groundwater discharge and other isotopic inferences and modeling studies.

Our findings suggest that the NWM accuracy would be improved by including agricultural return flows into the NWM water

sources. Agricultural return flows function as lower elevation baseflow fluxes, and are likely to be critical for sustaining stream-flow during drought periods or late in the growing season. Inclusion of this specific source into the more general groundwater fluxes would thus improve the ability to meet water manager and water user NWM data needs. Specifically, water managers use predictions of reach-specific flows at lower elevations during summer precipitation events and monsoons to assess flood risk, or to inform dam releases (if dam releases are incorporated into the NWM) to assess the volume of water required to achieve

specific management goals like fish species preservation or dam water level maintenance for hydropower production. Likewise, explicit inclusion of irrigation return flows in NWM calculations will assist in accurately predicting and projecting streamflows in heavily managed sections of river in the event of changing irrigation practices, increased evapotranspiration, or water supply reductions and fallowing of agricultural fields, which would change or halt irrigation groundwater fluxes. Finally, our findings have implications for areas at risk for diminished water availability due to issues of quality, arising from the entrainment of

fertilizer and pesticides and as well as dissolution and delivery of salts.

*Data availability.*  Data are publicly available (Reddy et al., 2023).

*Author contributions.*  Conceptualization and funding acquisition: ALP and OLM; Data curation: PCL, MCM, MK, JR, and JRB; Methodology, investigation and formal analysis: ALP and PCL; Project administration and supervision: ALP; Visualization: ALP; writing (original draft): ALP; Writing - review: OLM, PCL, MCM, JR, MK and JRB

*Competing interests.*  The authors declare no competing interests

*Acknowledgements.*  The authors acknowledge financial support from the US Geological Survey (USGS) Water Mission area Water Quality Processes Division. We appreciate comments from two USGS peer reviewers, whose comments improved the quality of the manuscript. The views expressed in this article are those of the author(s) and do not necessarily represent the views or the policies of the US Environmental Protection Agency. Any use of trade, firm, or product names is for descriptive purposes only and does not imply endorsement by the US

Government..





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
