# Peer review of "Isotopic evaluation of the National Water Model reveals missing agricultural irrigation contributions to streamflow across the western United States"

_Hydrology and Earth System Sciences, 2023_

## Author Comment (AC4)

**RC1**: ['Comment on hess-2023-308'](), Anonymous Referee #1, 25 Jan 2024

This paper presents an interesting work on the diagnose of NWM model with the aid of isotope data. The isotope simulation identifies different bias characteristics in high and low elevation regions. The bias of isotope simulation in low regions is attributed to the contributions of irrigation return flows, and some statistical analysis and literature reviews are conducted to support this hypothesis. Overall, the logic is clear, the analysis is solid, and the written is good, making this paper worth publishing in HESS. However, I would like to point out some major concerns related to the isotope dataset and mass balance calculation.

Thank you. We appreciate your comments and ideas.

1. The calculation of surface water isotope: From a hydrological viewpoint, the calculation surface water isotope ratio (equation 1) is confusing. The authors determine the isotope ratio through dividing the summed isotope fluxes by the summed runoff and groundwater fluxes. However, the term "runoff" usually refers to the sum of surface runoff and subsurface runoff. I don't know whether the "runoff" provided by NWM model refers to surface runoff or the total runoff. According to the equation, it seems that the runoff is actually only surface runoff. If this is the case, I suggest the authors to make it clear in the main text. Otherwise, the isotope ratio of surface water should be [Rgw*Fgw+Rp*(Fro-Fgw)]/Fro.

Thanks for this clarifying point. You're right – runoff in this case is only surface runoff, as it is defined and calculated that way by the NWM ( variable is 'runoff from terrain routing', qSfcLatRunoff, m3 s-1). The model also includes runoff from bottom of soil to bucket (qBtmVertRunoff, m3) and flux from gw bucket (qBucket, m3 s-1). See https://water.noaa.gov/about/output_file_contents for output file contents.

In the text I retained the more general definition of runoff in the introduction since it's more generally applicable but called out the potential for runoff to be from the surface or subsurface. In the methods section, specifically where I first introduce the NWM variables we use, I have added clarification to the definitions of the NWM variables. Later on, when the mass balance calculation is introduced, I clarified that the runoff is surface runoff. Throughout the results and discussion, I've updated 'runoff' to 'surface runoff' to help retain the distinction in the minds of the readers.

2. Choice of isotope dataset:

The authors adopted the monthly long term average precipitation and groundwater isotope data as the input data. This is okay for groundwater because its isotope composition is rather stable. However, the precipitation isotope usually has very strong

temporal variation, especially during wet season. Given that a high-resolution dataset of hydrological fluxes produced by NWM was adopted, it might be better to use a high-resolution precipitation isotope dataset, such as the output of isotope enabled general circulation models (iGCM, such as https://agupubs.onlinelibrary.wiley.com/doi/full/10.1029/2008JD010074). The reliability of the precipitation and groundwater isotope dataset itself also need to be evaluated, at least citing some descriptions about their accuracies in published papers. Otherwise, it would be hard to determine whether the simulation biases come from the NWM data or the isotope input data.

Besides, I find that the authors evaluated the isotope simulation performance by comparing the long term simulated results with the measurement surface water isotope sampled at a specific time. Such kind of measurement data would be highly dependent on the specific precipitation events and the corresponding isotope composition before sampling. So it might be more reasonable to compare the measurement data with the corresponding simulated result at the sampling time. We can observe a smaller range of simulated isotope compared to measurement in Figure 3. I think the aggregation of precipitation input and simulated isotope could be the reason of this.

Nonetheless, I understand that replacing the input data and repeating the whole calculation process is challenging. If this is difficult to achieve, please consider addressing these issues in a discussion section.

The authors agree that the differences in the time integrations represented by the model data and the observations are a limitation of this study, and also agree that future studies should consider higher temporal variability in the precipitation input value to help address temporal variability in observations arising from recent precipitation inputs to rivers.

Unfortunately, it is outside the capacity of the authors to re-do the study at a finer temporal scale at this point. However, we have plans to address this issue in future regional studies and fully agree that a more accurate treatment of the temporal component of the issue is a critical part of pushing this kind of study forward and improving our ability to evaluate water models using tracers.

To that end, thank you for the reference to the iGCM data – this dataset, or another precipitation isotope-reanalysis approach (maybe with WRF+NADP datasets), or a statistical approach like Finkebeiner et al., 2021 (https://doi.org/10.1175/JHM-D-20-0142.1) could be an excellent method to deploy during a second stage (higher

resolution, smaller region studies) of this project. We will take these ideas under consideration as we propose and develop continuations of this project.

Nonetheless, we have taken your suggestion to highlight the accuracies of the isoscapes we did use in the analysis. Considerations have been included in the methods section. Likewise, we have adjusted in the presentation of the evaluation of variability in the observation-model differences, choosing to highlight the strength of our dataset – spatial variability - and present the evaluation of interannual variability as insurance that the spatial variability doesn't co-vary with temporal variability due to interannual variability in sampling patterns. Hopefully this helps clarify for readers the strengths and weaknesses of this specific approach.

Specific issues:

- L142: Provide the full name of HUC2 at the place it appears for the first time.

  Added.

- L208~L215: How were the 10 random draws generated, and how were they used in specify to evaluate uncertainties? There seems to be results related to uncertainty in the result section.

  Both the groundwater and precipitation 'isoscapes' have uncertainty layers available. The uncertainty estimates, alongside the mean estimates from these products were leveraged for the uncertainty analysis. The random draws assumed a normal distribution of variance around the mean estimate. I have reworded this section, hopefully clarifying our approach.

- L400: Figure S8 should be S10?

  Thank you, good catch. The SI figures were in the wrong order so the reference in the main text has remained the same, but the order of SI figures has been updated.

- L442-443: This is a strong statement and is the basic of following analysis. Consider providing more explanation on it.

  We have added a few sentences of explanation to help readers follow our logic.

- L449: There is an additional letter "d"

  Fixed. Also fixed the superscripting of 18 in the same line.

- Please provide the r2 and p values in the scatter figures such as Figure 8 and 10

  Adding an r2 and p value isn't appropriate for Figures 8 and 10 because they aren't intended to show a linear relationship between two variables. They are intended to show non-linear relationships between multiple variables (i.e., $d_{diff}$ to elevation, stream order, aridity/climate and $d_{diff}$ to elevation, and water use characteristics of upstream area). Adding the r2 and p values are likely to confuse readers as to the focus of the figures. The linear regression statistics for other scatterplots (i.e., Figure 3, Figures S4-7) are available from tables in the text, which are now referenced in the figure captions. I also updated Table 1 to include the R2 of the regressions.

- Table 3: There is an additional symbol "+" in the last row

  Thank you, the extra + sign has been removed.

---

## Author Comment (AC5)

**RC2**: ['Comment on hess-2023-308'](), Anonymous Referee #2, 02 Feb 2024

Putman et al. use long term mean summer hydrology, gridded precipitation and groundwater isotope ratios, and in-stream water isotope ratio observations to evaluate the accuracy of the National Water Model, which is known to perform poorly at low elevations and in highly managed basins in the western United States. The authors use a water isotope mass balance approach to estimate river reach isotope ratios using National Water Model derived fluxes and compare 'modeled' isotope ratios with over 4,000 in-stream isotope observations. The differences between observed and modeled $\delta^{18}O$ and d-excess are used to evaluate statistical patterns in evapoconcentration of observations relative to modeled isotope values. Putman et al. conclude that offset between modeled and observed water isotope values are diagnostic of the lack of agricultural irrigation practices represented in the National Water Model and test this using 2015 Water Use Census data (annual water use by county) and developing an approach for estimating the amount of streamflow sourced from agricultural irrigation on a coarse catchment scale.

Overall, this manuscript is well written and provides a robust framework for evaluating model accuracy, considering different temporal and spatial patterns in the residuals, representing uncertainty in previously published datasets, and the utility of stable water isotope ratios in diagnosing misrepresentation of physical processes in hydrological models. This study is a solid contribution and worthy of publication in HESS; however, there are a few points I think need some clarification before the manuscript is finalized. Below is a discussion of my major comments, followed by more minor, line specific comments. Thank you for the opportunity to engage with this interesting and useful study.

Thank you. We appreciate your helpful review.

**Major Comments**

1. I'd like some additional discussion or clarity on why the interpretive framework in Figure 2 is applicable to the results of the d-excess(diff) and $\delta^{18}O$(diff) calculations presented. Typically, d-excess is calculated from the isotope ratios of a single/discrete water sample, but here d(diff) is calculated from isotope values that are a mix of spatially and temporally averaged modeled isotope ratios and point observations of in-stream isotope values. The framework in Figure 2 is based on dual isotope fractionation processes as units of water moves through the hydrological system. Given that the d-excess(diff) calculation here is based on spatially and temporally averaged modeled values, then d-excess(diff) and $\delta^{18}O$(diff), used as indices of evapoconcentration, are likely masking or missing a lot of variability in climate

conditions over time and space. Some additional reasoning would be useful for the reader. The discussion section describes how interannual variability is considered by looking at the regressions between the modeled values and the entire observational dataset versus the averaged observations, but I didn't clearly understand the lines being drawn between that interannual variability (due to variable climates?) and the consideration of how much variance is missing in the NWM that can be attributed to agricultural return flows.

I'm trying to parse the question of the reviewer, which I'm not totally sure if I understand, so forgive me if the explanation begins a little too simplistically. I don't want to make any logic leaps. Hopefully I satisfactorily address your concern in the paragraphs below.

The idea behind this framework is that it helps us interpret the meaning of the deviations of the observations from the estimates based on the mass balance approach. If the model and data inputs correctly capture all isotope-influencing sources and processes, then all points would cluster around (0,0). Instead, we see a spread along an 8:1 line as well as deviations from that 8:1 line. The structure of the deviations from the 8:1 line indicates that negative d18O(diff) tends to be associated with positive d(diff) if d(diff) is non-zero, whereas positive d18O(diff) tends to be associated with negative d(diff) if d(diff) is non-zero. This structure arises in this case because the mass balance approach tends to produce estimates with d-excess values of close to 10 (see Figure 3, gray dots), indicating no evidence of non-equilibrium processes influencing river processes (e.g., evapoconcentration (- d), mixed phase cloud processes (+d), snowmelt fractionation (+d)). On the other hand, observations have a wide range of d-excess values, though most tend to be close to or less than 10 (see Figure 3, blue dots), which are characteristic of evapoconcentration, though some plot above the GMWL, indicating potential for condensation-oriented non-equilibrium processes (e.g., snow processes). So, when the two datasets are compared / differenced, the non-equilibrium signals in the observations are highlighted.

So, the interpretation framework is predicated on the (unintentional) fulfillment of an equilibrium assumption by the mass balance approach and the deviation from that assumption by the observations. If the mass balance approach yielded some evidence of non-equilibrium signals (due to input data or process), then the interpretation framework would likely have different implications. This logic is already largely in place (in brief) in the methods:

"We can interpret combinations of d18O(diff) and d(diff) together, as well as d(diff) independently to infer the uncharacterized sources responsible for the observation-model difference. This framework is useful because the ratios of d2H to d18O of the

isotopic inputs to the isotope mass balance tend to be close to 8, whereas those from the observations more often differ from 8. This means that all non-zero d(diff) values can be used to identify omitted water sources and where they are important to streamflow." However, I have added a caveat that the interpretations of the framework would change if the characteristics of the null hypothesis change (i.e., don't represent equilibrium conditions/an equilibrium assumption).

As for attribution of the source of the evapoconcentration signal, the reviewer is correct in that there are many factors that can influence the deviation of the observation from the model. We attempted to evaluate the potential influence of interannual and seasonal variability as explanations for the signal. Certainly, both of those modes of variability are responsible for some scatter in the results, as we demonstrate in the different sections of the manuscript. However, among the modes of variability we evaluated, the spatial variability was the most consistent across spatial domains and remained even when using average values. Unfortunately, due to the nature of our approach, it was not possible to evaluate all three modes of variability simultaneously, especially because of the sometimes small number of high leverage points, so the evaluation of the spatial mode certainly includes scatter from the interannual and seasonal modes of variability as well as other, unevaluated sources of variability. This likelihood of scatter from other sources of variability probably accounts for the lower predictive power of the statistical approach. However, to avoid overfitting our model, particularly in basins with fewer observations, we did not attempt to statistically evaluate all identified modes of variability simultaneously. I added a caveat specifically calling out the inability to directly address temporal aspects of variability in the methods section, and a nod to the contribution of temporal variability as a cause for scatter / low variance explained in the discussion section.

Evaluating all modes of variability at once might be possible in future studies that are able to resolve the temporal aspect of variability (i.e., producing estimates for each month and year) to match observations, and for smaller scale studies with higher temporal resolution sampling and input data. We hope that we, or others may be able to pursue this approach as an improvement to what we've put forth in this initial study.

2. The diagnosis of the National Water Model inaccurately representing agricultural return flows is well reasoned in the study and a conclusion that makes sense given the difficulty of many hydrological models in representing irrigation practices given that water use data is difficult to obtain (water users are often reluctant to share this information freely in highly managed areas). One concern I have is the practicality of reasonably incorporating agricultural return fluxes into the model and the approach for estimating this contribution taken in the manuscript. The simplified way of calculating ratios of water use contributions to stream flow in this study seems like a

reasonable first pass, but there are many unknowns. For one thing, water use can vary widely between water year types – so including some level of uncertainty or variability in the 2015 Water Use Census data would be helpful. For example, 2015 was a critically dry year in California, so water use data is likely reduced during that year compared to a wetter year on record and the ratio of groundwater to surface water use in the Central Valley is likely inflated for that year. Applying water use data that is from a specific year as a point of understanding contributions to streamflow should likely consider the isotope data from that specific year to match, since it's not representative of long-term mean conditions. I'd like to see some explicit discussion of what the water use data represents in the main text and whether it's representative of long-term conditions. One thing that could be considered (it may or may not be appropriate here) is the EPA's EnviroAtlas' dataset of different types of water use. They have estimated longer term datasets for agricultural water use, industrial, domestic, etc. https://www.epa.gov/enviroatlas

Thank you for this consideration. We agree – the use of annual-scale water use and the general approach taken could certainly lead to errors in our evaluations due to the potential for oversimplification, particularly because our observational data may come from anytime between 2000 and 2021.

Fortunately, the USGS has *just* released month and HUC12-scale estimates of water use (Haynes, et al., 2023, Monthly crop irrigation withdrawals and efficiencies by HUC12 watershed for years 2000-2020 within the conterminous United States: U.S. Geological Survey data release, https://doi.org/10.5066/P9LGISUM). We will evaluate the feasibility of replacing the analysis using the 2015 data with one using this improved dataset to address issues with uncertainty in this analysis.

If replacing the water use data product with the higher resolution version is not feasible, we will re-run the analysis using the mean of the 2000, 2005, 2010, and 2015 water use datasets. We would use these years since they are what is currently available at the same spatial scale as our original analysis.

**Line Specific Comments**

Line 89-92: d-excess is typically calculated for corresponding $\delta^2H$ and $\delta^{18}O$ values for a specific sample/observation. In this study, d(diff) is calculated from estimated average isotope values. Is d-excess still a reliable metric for evaporation when you are calculating from long term averages/values calculated using mass balance? It seems like the mass balance calculation step would not include all the non-equilibrium fractionation processes that could impact the d-excess value. Some additional reasoning somewhere in the text would be helpful for the reader.

This line-specific comment is related to the general comment on the framework discussed above. In this case, yes, it is. However, that's because the mass balance approach produces results that are effectively a null hypothesis that reflects only equilibrium conditions (i.e., d2H/d18O ratios of about 8) whereas observations vary more widely reflecting influences of non-equilibrium processes on different source waters. As mentioned in general comment 1, I have added a caveat in the description of the framework that in cases where a modeled value contains some non-equilibrium signal (d2H/d18O ratios different than 8), the interpretations of the d(diff) values may be different. I have also added "The model estimates reflect an assumption that water sources contributing to streamflow were subject only to equilibrium fraction, whereas observations indicate contributions of waters influenced by non-equilibrium processes." to the results section "Model-observation differences" to help clarify the applicability of d(diff) to diagnosing water sources with non-equilibrium conditions.

Line 108-110: remove "associated with irrigation"

Removed.

Figure 1 Caption: in text citation format typo for (Bowen 2022b)

Fixed.

Line 191: "Where available, we filled these data gaps using method outlined in Text S2." I would briefly explain that the authors used the gridded DJF precipitation isotope products to help fill the gaps, since they are listed in Figure 1, but not mentioned anywhere in the main text.

I have updated the sentence to "Where available, we filled these data gaps using either other groundwater depths or mean winter precipitation (DJF) as described in Text S2"

Line 220: typo, should be "This decision was made…"

Fixed.

Line 245: "We evaluated the results with all unaveraged observations and mean isotope ratio at river reaches with multiple observations." I'm not clear on what this means. The correlation/regression analyses would need to be done between monthly average isotope ratios for an apples-to-apples comparison, rather than mixing discrete observations with monthly average model values.

I believe this misunderstanding reflects some confusion about our approach. The mass balance results are available at the long-term average summer (JJA) season scale (not

long term average monthly or year-month average scale). The comparisons are made to all data points (unaveraged) and with mean values for reaches with more than one observation (averaged).

We include the comparison results with all data (even though we acknowledge that the timescale of the observation and the modeled result are different) to evaluate the results at a greater number of reaches and thus covering more of the spatial domain. However, that approach leads to scatter due to mismatches in the time integration of the modeled vs observational data, so we also included a comparison using only reaches with more than one observation using the average observational value at that reach.

I haven't made a change to the text as I believe there is sufficient information available that the description is clear and the issue was not flagged by other reviewers.

Line 250: "( Text S3)" has an extra space after first parenthesis.

Fixed.

Figures 6: This is a really nice figure illustrating the different temporal evolution of $\delta^{18}O$(diff) and d(diff) throughout the different major basins in the western US! Please list the distribution statistics in the caption (i.e., box represents what percentiles, what are the smaller, shaded boxes in the Great Basin boxes, diamonds are outliers?).

These kinds of plots are called 'letter-value plots' or in python, 'boxenplots' (Hofmann et al., 2017, https://doi.org/10.1080/10618600.2017.1305277). After further consideration, I think that this plot would be better off as a boxplot instead, so will update the plot and added the standard boxplot description to the caption so the meaning of the domain is clear.

Line 454: typo, remove "a" after due to

Removed.

Line 458: $p<0.001$ is listed for significance, whereas it has been $p<0.1$ or $p<0.01$ in other parts of the manuscript. I suggest staying consistent with listing the p-value in the text.

Figure 8: y-axis label I believe should be listed as ‰ instead of %.

You are correct. It has been changed.

Line 468: The meaning of the first sentence is unclear. The $\delta^{18}$O(diff) and d(diff) are statistically significant relative to what?

Yes, this sentence is not clear. We meant to say that the values of d18O(diff) and d(diff) values in headwater areas are statistically different from 0. The sentence has been revised for clarity.

Figure 9: Please explain the statistics of what the boxplots represent in the caption. Also, is not all data shown? Every land cover type looks like it has groundwater d less than -20 but that's where the plot stops.

These kinds of plots are called 'letter-value plots' or in python, 'boxenplots' (Hofmann et al., 2017, https://doi.org/10.1080/10618600.2017.1305277). These kinds of plots are useful for datasets with 1) a large amount of data and 2) show more detail about data distribution than boxplots. They are used in this case because the data are numerous and have a non-normal distribution. Letter-value plots (boxenplots) start with the median (Q2, 50th percentile) as the centerline. Each successive level outward contains half of the remaining data. So the first two sections out from the centerline contain 50% of the data. After that, the next two sections contain 25% of the data. This continues until the outlier level. The plot is cut off at -20‰ because more than 85% (and up to 95%) of the data is displayed on the plot, but the tails are quite long. The point of the figure is effectively made with the median and 75% of the data.

I have added more description of the plot type to the figure so it's clearer to the reader.

Line 556: typo, should be "stream"

Fixed.

---

## Author Comment (AC6)

**RC3**: 'Comment on hess-2023-308', Anonymous Referee #3, 06 Feb 2024

This is a well-written, comprehensive study that applies a novel evaluation framework using isotopes to diagnose potential error sources in hydrologic models. The study offers useful analyses and conclusions for hydrologic model evaluation, with a focus on processes, which are supported by multiple lines of evidence. I believe the study merits publication, but I have several main comments that I offer for the authors to consider:

Thank you for your helpful review!

- **Isotope tracer expertise is assumed**. I suspect that many HESS readers are not experts in isotope tracers, and there is a lot of assumed expertise and jargon. The manuscript would benefit from more explanations up front and throughout so that readers unfamiliar can benefit from this novel approach. I point out examples in my specific comments.

  Based on your suggestions, I've made substantial changes to the way I introduce and describe isotope systematics in the paper, with an eye to how a non-expert may read the work. I hope these changes are effective and improve reader clarity and interest.

- **The main conceptual method is hard to pick out from all the details**. The authors do an excellent job of explaining their methods in great detail, but I found myself missing the forest for the trees in my first read through. Having never read a tracer study, the main approach – comparing the observed isotopes with the NWM derived estimates (which comes from both NWM fluxes AND gridded datasets) was hard to follow. I think this should be clarified earlier, and I offer a few suggestions to improve the Method section organization in my minor comments. For example, Figure 1 is very detailed, but hard to follow all that's going on at the outset of the Methods, so I suggest adding a general conceptual overview and/or simple flow chart to guide the reader (including what's in Equation 1, otherwise it appears much too late for the reader to follow what's happening). I recognize that this is a subjective suggestion, but I think it would help to increase the reach of your paper.

  Thank you for this feedback. Based on your suggestions, we have attempted to clarify the approach in the abstract, intro, and methods. Based on your comments, it seems that much of the confusion can be cleared up by emphasizing the observation-model comparison nature of the study first in all places, and following up with the specifics around producing the 'modeled' river isotope data. This emphasis has been applied throughout and is described in response to specific comments.

- **The key results are hard to pick out from the supporting results**. I appreciate the comprehensive results and multiple lines of evidence presented, but will caution the authors that it can make it difficult for readers to focus on the key results on which the main conclusions are based. The authors may want to review all the results and see if there are any that they might like to include in the Supplemental (I make a few suggestions). I recognize that this is a subjective suggestion, and that there is a tradeoff to including too few versus too many results, but I think a slightly more curated results section would help to increase the reach of your paper. (Also, I will note that the authors do a nice job of summarizing the key results in the Abstract and Conclusions, so this is just a suggestion for the Results themselves).

  I am hopeful that our re-arrangement of the presentation of the latter part of the results helps with this challenge in extracting the main results from the large amount of information presented. At your suggestion, I removed Section 3.3, which covered interannual variability to the supplemental. I then swapped Section 3.4 and 3.5 so that the spatial analysis comes first, and the seasonal/growing season evolution section comes second and can used as support for the conclusions of Section 3.5 (now 3.3). Hopefully this highlights the main objective of the paper, which is about agricultural return flows, rather than assessing all modes of variability in the dataset.

**Specific comments**:

Abstract:

Line 3: Is "fidelity" the right work here?

The Merriam-Webster dictionary defines fidelity as "the quality or state of being faithful or loyal", "exactness in details", and "the degree to which an electronic device (as a record player, radio, or television) correctly reproduces its effect (as sound or a picture)". The word is present in reference to models in a similar way in which we've used it in such papers as:

- "Fidelity of WRF model in simulating heat wave events over India" (https://www.nature.com/articles/s41598-024-52541-2)
- "Advancing Process Representation in Hydrological Models: Integrating New Concepts, Knowledge, and Data" (see first line of abstract, https://doi.org/10.1029/2021WR030661)

- "Learning from hydrological models' challenges: A case study from the Nelson basin model intercomparison project" (see abstract, https://doi.org/10.1016/j.jhydrol.2023.129820)

I believe our usage is consistent with other descriptions of model performance elsewhere in the literature, so will leave it as is.

Line 4: In parenthesis, I am not familiar with the delta 18 Oxygen and delta 2 hydrogen notation. Is there a way to define them here for non-experts? Maybe just remove the parenthetical all together from this sentence? You may need to define them later (line 8, etc).

I recognize that the water isotope notation was not introduced yet when these were referenced. The idea here is that skipping the notation may be ok in the abstract, as long as the full definition is produced as soon as the information appears in the introduction. This is typically done (it occurs in other water isotope-focused and tracer-focused publications in general journals) because the abstract is too short to allow for a full definition, yet the symbology is included to stimulate reader recognition and clarity about which isotope systems will be discussed. I changed the initial parenthetical statement slightly to indicate the isotope ratios we are using and mention that they are expressed in delta notation.

Line 4-7: I had a hard time understanding generally what you did from these sentences until I read the manuscript and thoroughly studied Figure 1. Here's a suggestion of what might help a reader like me (at least in terms of laying out the general conceptual framework of what was done – please fix details if I have them wrong):

"In this study, we compare observational river isotope data with estimates of river isotopes derived from the NWM. The evaluation is done in 5 basins in the western US in summer from 2000 and 2019. In terms of observations, we use 4503 in-stream water isotope observations in 877 reaches. In terms of the corresponding estimates of river isotopes, these are calculate using a mass balance equation based on NWM-fluxes and estimates of isotope ratios from long term mean gridded precipitation and groundwater datasets."

I have made some changes to the abstract wording that are similar to those suggested.

Introduction

Line 18-26. This paragraph is quite general in scope and I don't think is needed. I think you could delete and start with paragraph 2 (i.e., line 27), using something like the first line of your abstract, i.e., : "Hydrologic models, such as the National Oceanic and

Atmospheric Administration's National Water Model (NWM), provides critical analyses and predictions of streamflow that support water management decisions. The NWM is an application of the WRF-Hydro model (Gochis et al., 2018), and is fully routed with high spatial and temporal resolution, providing short and medium term streamflow ... " etc.

The initial paragraph lays out some important motivation for why accurate streamflow estimates are important, and why pursuing improving accuracy of operational hydrologic models is crucial. I plan to retain the paragraph.

Line 27: The operational NWM is based on the WRF-Hydro model (not its data); the NWM is an application of the WRF-Hydro model. Perhaps you are confusing WRF-Hydro (a hydro model) with WRF (a meteorological model). Suggest saying, "...like the National Oceanic and Atmospheric Administration's National Water Model (NWM) which is an application of the WRF-Hydro model (Gochis et al., 2018)" or you could say "is based on the WRF-Hydro model (Gochis et al., 2018)."

Thank you for this correction. We have updated the text.

Line 34: "fidelity" does not seem like the right word here.

As above, I believe our usage is consistent with other descriptions of model performance elsewhere in the literature, so will leave it as is.

Line 84: Sentence that starts with "tracers" – can you add a general sentence on tracers and isotopes (before getting into the 16O, etc), for the non-expert? When you get to the parenthetical 16 oxygen, etc, please define these. I start catching your drift a bit later when you define 2H/18O as "heavy" and 1H and 16O as "light", though as a non-expert I'm not sure to what it is compared.

This paragraph has been updated, including additional clarity around the concept of tracers, with general examples of tracer studies.

Line 89: The expression: "The secondary parameter, deuterium excess.." is not clear to a non-expert.

I have expanded this section and hopefully with the changes to the whole section, is clearer to a non-expert reader.

This paragraph (lines 84-92) is very dense, especially for a non-expert (of isotopes), try rereading as someone who doesn't know about isotopes and generalizing a bit as possible, to help the reader understand this powerful evaluation tool.

I have made substantial changes to this paragraph to help lead non-experts through the basics of isotope systematics, while have also made an effort to keep the paragraph condensed for clarity and readability of the entirety of the introduction. Hopefully the clarifications help.

Line 93-94. What is isotopic fractionation? Can you say this another way for the non-expert?

I have updated this section to add a cursory definition of fractionation.

Line 110: What is evopoconcentrated and evaporative enrichment?

I have updated these to more general descriptive language.

Line 115-116: Same as in the Abstract, I wasn't really sure what you did until I read through and studied Figure 1, although this is easier for me to understand than the Abstract. Could you start most generally, saying: "In this study, we compared stream water isotope observations with estimates of water isotopes derived from an isotope mass balance. The isotope mass balance is from xyz NWM and lmnop gridded long term, etc".

I flipped the first and second sentences to highlight the comparison aspect over the mass balance aspect, as you suggested, and made minor editorial changes to the sentences for clarity.

Methods

127-128. Same as previous comments, I initially had trouble understanding what was done here. Even though I like Figure 1, it is very dense. I'm wondering if you could have a very simple conceptual figure first to ground the reader before Figure 1, where you just show the 3 main pieces: (1) Direct obs, (2a) NWM, (2b) gridded ratios, as well as equation 1 and where the pieces fit in on the model side. This could be in a section called "2.1 Conceptual Framework", and could include the general figure, which would introduce sections 2.4 and 2.5, and it would absorb the current specs listed in "2.1 Temporal domain", "2.2 Spatial..." and "2.3 Data assim". Then Figure 1, with all the details, could come later.

My intent was for the first paragraph of the methods to function as the 'conceptual framework'. Following other suggestions by the reviewer, I altered the text in this initial paragraph to highlight the comparison part of the work first, and follow with the description of the modeling approach. I also simplified some of the methods so they might be more clear, allowing a reader to absorb the conceptual framework before

diving into the methods. I think an addition, simplified conceptual framework figure is not necessary with the improvements to the methods section and text throughout.

Figure 1: This is a very nice figure, but dense... See previous comments. One note: For NWM data feeding into Equation 1: maybe have the notation (Fgw) and (Fro) and for obs data have (Rgw) and (Rro) there to link with Equation 1. This might be better suited to a more general conceptual figure though (see previous comment).

We have updated this figure to include that notation to help link the conceptual figure to the equation and the overall methodological concept.

2.3. Data assimilation: This seems minor to be a full section, and I wasn't sure what this was related to. Can this be part of 2.2 Spatial domain or just Supplemental? Or if you decide to have a Conceptual Framework section, it could be absorbed in that.

I absorbed this section into the Spatial Domain section, as suggested, and modified the text for simplicity and clarity.

Line 160. Correction, the NWM is based on the WRF-Hydro model – which is an open source, community hydrologic model, it is not based on inputs from it (I think you might be confusing WRF-Hydro with WRF, where WRF would provide inputs): "The operational hydrologic model is based on the open-source, community hydrologic model, WRF-Hydro (Gochis et al., 2020b, a)..."

Thanks for that clarification. Text has been updated.

Line 170: Do you mean Figure 1 here?

Yes, that's correct. Thank you for the catch.

Equation 1: Seeing this equation helped me to see how the pieces fit together. If you decide to include a conceptual model, I suggest having this equation in it to see how each piece fits in (you could do it generally, for just one reach, as a demonstration, so it was a simpler equation without the subscripts).

Thank you for this suggestion and insight.

Figure 2 – I like this figure and how it showed the way to interpret. I often had to look back at this figure to interpret later results.

Thank you. That was the intent of the figure – to help with interpretation of the results.

Line 371: What is a meteoric water line? What is a surface water line?

A description of WLs is now in the introduction, which should help clear up this section.

Line 380: Is Table 1 needed for the main text or could it go into the supplement?

We will retain it in the main text.

Line 386. What is an isotopologue?

Isotopologue has now been introduced in the introduction, which hopefully clears up this comment.

Line 405: "The strongest signal in our data is that of evaporation, evidenced by combinations of positive $\delta^{18}O_{diff}$ and negative $d_{diff}$ in arid regions." <- This is an important conclusion, but there are so many results it's hard to quickly see what evidence this is from – I had to really go back and study all the figures and tables to realize I needed to imagine all the points from Figure 4 as if they were on Figure 2. Can you add something to that effect to guide the reader? Or add the colored quadrants to remind the reader? Or maybe just say in the caption of Figure 4, "see Figure 2 for what the different locations on the x- y- axis mean"?

I added a sentence to the caption of Figure 4 that refers the reader to Figure 2 for interpretation of the scatter plot. I think adding annotation to the plot would make it too busy and would make Figure 2 redundant.

Table 2: Is this need for the main text or could it go into the supplement?

We will retain it in the main text.

Section 3.3. This section and Figure 5 did not seem particularly important to the results/conclusions, and one suggestion would be to put it in the Supplemental (so that the other key results are less buried). If so, you could just have one sentence at the end of the previous section saying something like "There was little interannual variability, which we interpret to mean there was pervasive presence of eval… etc.. see Supplemental xxx".

I took your advice and put this section and figure into the supplemental data. I also moved Section 3.4, which was concerned with temporal evolution of stream waters over the growing season in different basins and used that as support for Section 3.5, which is the primary result of the manuscript. I updated the transition paragraph to

reference the interannual variability analysis in the SI as a basis for the robustness of the spatial analysis in the context of the dataset.

---

## Author Response (AR2)

Minor revisions of HESS-2023-308: Response to reviewers

Reviewer 1: Many thanks to the authors for the revision. Most of my concerns have been addressed in the revised manuscript. The authors have clarified the runoff concept and highlighted the accuracies of the isoscapes.

Thanks for taking the time to re-review the manuscript!

However, considering that there is still room for the improvement of this work in terms of the temporal scale and dataset choice, just as mentioned in the response letter, I would like to suggest authors to address these limitations and future directions in the main text by 2~3 sentences.

Good point. Since the manuscript doesn't have a specific 'limitations' section, I've included these ideas as discussion points at the end of section 3.1, where the observation-model differences are presented and discussed, and the end of section 3.3, where the limitations of the statistical analysis relating irrigation to the isotopic response are discussed and finally in the conclusions section.

One minor issue: there is only one sub-section 3.4.1 in the section 3.4. Please check it.

Good catch. I have adjusted the sections so section 3.4 is generally about additional lines of evidence supporting the main finding from section 4, with subsections 3.4.1 and 3.4.2 which detail evidence from our data and from other literature and data sources, respectively.
* * *
Reviewer 2: A minor revision is needed to the text to reflect a change the authors made in the data used for the water use evaluation.

I checked the text thoroughly and am not sure exactly where this change needs to be made. The references to the data used for the water use analysis are general and accurate in the main text and only specific in the SI. Everything in the SI has been updated to the correct values. I am wondering if the reviewer is referring to the references I make to the 2015 water use dataset in the introduction? These references simply discuss general patterns in water use in the basins (i.e., 80% of water use in western states is for irrigation). It is appropriate to keep those references to the 2015 water use report, even though we use more data in our analysis.
* * *
Reviewer 3: No comments.